# Purkinje cell axonal swellings enhance action potential fidelity and cerebellar function

Daneck Lang-Ouellette [1], Kim M. Gruver [1,2,5], Amy Smith-Dijak [1,5], François G. C. Blot[3], Chloe A. Stewart[1,2], Pauline de Vanssay de Blavous [1], Connie H. Li [1], Carter Van Eitrem [1], Charlotte Rosen [1], Phyllis L. Faust [4], Martijn Schonewille [3] & Alanna J. Watt [1✉]

Axonal plasticity allows neurons to control their output, which critically determines the flow of information in the brain. Axon diameter can be regulated by activity, yet how morphological changes in an axon impact its function remains poorly understood. Axonal swellings have been found on Purkinje cell axons in the cerebellum both in healthy development and in neurodegenerative diseases, and computational models predicts that axonal swellings impair axonal function. Here we report that in young Purkinje cells, axons with swellings propagated action potentials with higher fidelity than those without, and that axonal swellings form when axonal failures are high. Furthermore, we observed that healthy young adult mice with more axonal swellings learn better on cerebellar-related tasks than mice with fewer swellings. Our findings suggest that axonal swellings underlie a form of axonal plasticity that optimizes the fidelity of action potential propagation in axons, resulting in enhanced learning.

[1] Biology Department, McGill University, Montreal, QC, Canada. [2] Integrated Neuroscience Program, McGill University, Montreal, QC, Canada. [3] Erasmus University Medical Center, Rotterdam, Netherlands. [4] Department of Pathology and Cell Biology, Columbia University, New York, NY, USA. [5] These authors contributed equally: Kim M. Gruver, Amy Smith-Dijak. ✉email: alanna.watt@mcgill.ca

Information is transmitted in the nervous system primarily by action potentials traveling along axons. This means that a neuron's ability to maintain high-fidelity axonal propagation is of fundamental importance for its function[1]. Indeed, instances when axonal propagation is delayed or interrupted can produce devastating consequences. For example, in multiple sclerosis (MS), axonal impairments resulting from the breakdown of myelin that surrounds axons lead to severe sensory and motor symptoms. Conversely, alterations in axonal structure can also be adaptive: for example, neurons respond to elevated levels of activity by restructuring their axon initial segment (AIS) to homeostatically regulate their excitability[2].

Since Purkinje cell axons carry information out of the cerebellar cortex, changes in the structure of their axons could impact cerebellar function dramatically. Purkinje cell axonal swellings appear transiently during cerebellar development[3,4] and are observed during normal aging[5], including in healthy human samples[6,7]. These data suggest that axonal swellings play a physiological role in the brain. However, similar axonal swellings have also been associated with axon dysfunction in several neurodegenerative diseases[8–11], indicating that at least in some cases, axonal swellings may be implicated in pathological function. Whether axonal swellings serve different roles in different contexts remains to be determined. Computational modeling has proposed that axonal swellings serve a pathophysiological role, as models predict that action potentials will be delayed, filtered, or fail when propagating across an axonal swelling[12–14].

To determine the impact of axonal swellings on Purkinje cell axonal function, we performed visually-targeted dual recordings from the soma and axon of individual Purkinje cells from young mice, and found that Purkinje cell axonal failures were reduced in axons with swellings. Axonal propagation was more reliable in axons with swellings when axon failures were induced by driving the neuron to fire at near-maximal firing rates. Pharmacologically mimicking high levels of axonal failures led to the formation of focal axonal swellings, and we uncovered that their formation is $Ca^{2+}$-dependent and requires $Ca^{2+}$ entry through voltage-gated $Ca^{2+}$ channels. Finally, by examining cerebellar-related behavior, we observed that mice exhibiting higher levels of cerebellar learning had higher numbers of axonal swellings. These data suggest that the enhancement of action potential propagation associated with axonal swellings in healthy young animals positively impacts behavior.

## Results

**Action potential propagation is enhanced in axons with swellings.** Purkinje cell axonal swellings are present in healthy developing rodents[4] and are also observed in several neurodegenerative diseases[8–11]. However, functional measurements of the impact of axonal swellings on axons have been lacking. We used dual targeted loose-patch recordings with fluorescently-tagged Quantum dot-coated glass electrodes[15] to measure action potentials simultaneously in Purkinje cell somata and axons (Fig. 1a) in acute brain slices prepared from the cerebellar vermis of juvenile, healthy transgenic mice expressing GFP in Purkinje cells, including in their axons[16,17]. Since computational models suggest that axonal swellings increase axonal failure rates[13], we monitored action potential failures in Purkinje cells with and without focal axonal swellings (Fig. 1b). Approximately 30% of axons display swellings at this age[4], and the vast majority of these were single swellings within the granule cell layer (98.7%; Supplementary Fig. 1), which we targeted for our recordings of axonal swellings. Axons without swellings were recorded as control axons at similar distances from the soma (Supplementary Fig. 2 and Supplementary Table 1). Surprisingly, we found that

the axonal failure rate in Purkinje cells with swellings was significantly lower than that in Purkinje cells without axonal swellings (Control Axon: $6.07 \pm 1.36$ per 1000 spike; $n = 11$; Axon with Swelling: $1.12 \pm 0.41$ per 1000 spike; $n = 9$; Mann–Whitney $U$-test, $P = 0.001$; Fig. 1c). A large proportion of axons with swellings propagated action potentials with very high fidelity (having very few axonal failures), whereas only the occasional axon without a swelling propagated with similar fidelity (9.1% of control axons were high-fidelity; 55.5% of axons with swellings were high-fidelity; Mann–Whitney $U$-test, $P = 0.05$; Fig. 1d). We found no differences in the axosomatic delay between recordings in axons with swelling and control axons (Fig. 1e and Supplementary Fig. 2), suggesting that swellings do not change the propagation speed of action potentials, or at least not over the relatively short distances that we have measured (Supplementary Fig. 2 and Supplementary Table 1). Taken together, these data argue that axonal swellings enhance rather than impair axonal propagation.

We wondered whether firing properties would differ in Purkinje cells with axonal swellings, since changes in Purkinje cell firing properties are observed in diseases where Purkinje cell axon swellings are observed[18–21]. We found that Purkinje cells with axonal swellings fired action potentials at rates that were indistinguishable from those with axons without swellings (Fig. 1f) and observed no changes in regularity (Supplementary Fig. 3). Furthermore, we observed no relationship between baseline firing rate and axonal failure rate for axons with or without swellings (Supplementary Fig. 3).

Axonal failures have been reported to be elevated when Purkinje cell firing is driven at high frequencies ($> \sim 250$ Hz)[22–25]. We thus wondered whether frequency-dependent failures would be affected by axonal swellings. To address this, we made whole-cell recordings from Purkinje cell somata combined with simultaneous extracellular recordings from the axon from Purkinje cells with axons without swellings (control axons), and from Purkinje cells with axonal swellings (Fig. 2a). We injected current in 0.05 nA steps through the somatic patch pipette to elicit firing over a range of frequencies in order to measure the axonal failure rate across frequencies in cells with and without axonal swellings (Fig. 2a). Control axons showed high axonal propagation rates at low frequencies (Fig. 2b), but axonal propagation fidelity decreased as firing frequency increased, (Fig. 2c, d), consistent with previous findings[22–25]. Axons with swellings displayed high axonal propagation rates across frequencies (Fig. 2b–d), with significantly higher propagation reliability at maximal firing rates (Fig. 2d). Consistent with previous reports[26], the Purkinje cells at this age have a range of maximal firing frequencies that tend to be lower than for mature Purkinje cells (Fig. 2d). Similar to our findings for spontaneous firing properties (Fig. 1e), we observed no differences in the axosomatic delays between action potentials (Fig. 2e). These data argue that the reduction of firing failures observed in axons with axonal swellings is even more pronounced at firing frequencies that result in higher levels of axonal failures in axons without swellings.

To examine axonal swellings in greater detail, we studied their ultrastructure by imaging anterior lobules of cerebellar vermis with transmission electron microscopy (TEM; from $N = 7$ mice). We identified 15 myelinated spheroid-shaped structures with smallest diameters greater than 4 μm as putative axonal swellings (Fig. 3a, b; swelling diameter average = 8.1 μm; range from 4.1 to 11.8 μm). We found no evidence of presynaptic or postsynaptic specializations, confirming that axonal swellings are not presynaptic terminals or specialized postsynaptic structure for axo-axonal synapses. Axonal swellings were myelinated and we fortuitously found two instances where an axonal swelling had

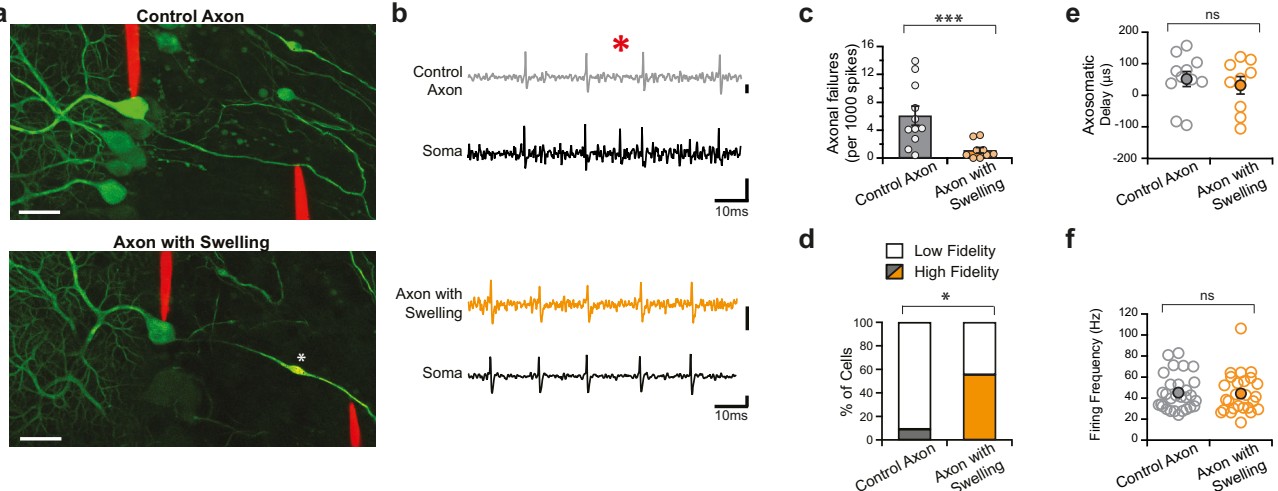

**Fig. 1 Axonal swellings are associated with improved axonal propagation of action potentials. a** Representative images of a Purkinje cell with placement of electrodes showing simultaneous dual recording from Purkinje cells soma (left electrode) and axon (right electrode) from a control axon (top) and an axon with a swelling (bottom; asterisk indicates swelling). Images were acquired after each simultaneous dual recording. Scale bar, 25 μm. **b** Representative traces of dual recordings from soma (black) and axon corresponding to Purkinje cell axons without (top, gray) and with (bottom, orange) an axonal swelling. Red asterisk indicates action potential failures in the axon. Scale bar: 0.5 mV for top two traces, 1 and 0.5 mV for bottom two traces, respectively. **c** Axonal spike failure rate is significantly reduced in axons with swellings (Control axon, $n = 11$; Axon with swelling, $n = 9$; significantly different, one-tailed Mann–Whitney $U$-test; $P = 0.001$). **d** Purkinje cell axons with swellings are more likely to be high fidelity transmitters (high fidelity: <1 of 1000 action potentials fail; low fidelity >1 in 1000 action potentials fail) than axons without swellings (two-tailed Mann–Whitney $U$-test; $P = 0.050$). **e** The axosomatic delay between spikes is not significantly different for control axons (gray) or axons with swellings (orange) (two-tailed Student's $t$-test, $P = 0.59$). Axosomatic delay can be negative when the action potential reaches the axonal recording electrode before the somatic electrode, since the spike initiation zone is located in the axon initial segment. **f** The spontaneous firing frequency of Purkinje cells with axonal swellings is not different from cells with control axons. (Firing Frequency: Control axon: 45.2 ± 3.15 Hz; $n = 29$; Axon with swelling: 44.3 ± 3.71 Hz; $n = 26$; not significantly different, two-tailed Student's $t$-test $P = 0.70$). Data were presented as mean ± SEM (***$P < 0.005$; *$P < 0.05$; ns $P > 0.05$). Source data are provided as a Source Data file.

intact flanking axonal segments (one example shown in Fig. 3a), which revealed that the myelin thickness surrounding the axonal swelling was similar to that around flanking axons (Supplementary Fig. 4). Occasionally, we observed perinodal protrusions proximal to an axonal swelling (Fig. 3a, white asterisk), which agrees with our findings that about half of axonal swellings are near the paranodal protein CASPR (contactin-associated protein 1; $n = 58$ axonal swellings; Supplementary Fig. 5), suggesting that swellings are frequently found close to paranodal junction, and therefore are proximal to nodes of Ranvier. Additionally, we occasionally observed periaxonal oligodendrocytic cytoplasm associated with axonal swellings (Fig. 3a cyan box inset, observed in ~20% of swellings), which has been previously described[27]. Although the function of specialized oligodendrocytic structures like perinodal protrusions is poorly understood, it suggests that axonal swellings may preferentially form close to them. Given that perinodal and internodal structures play a role in saltatory conduction[28], our results suggest that oligodendrocytic specialization and perinodal protrusions proximal to axonal swellings may contribute to enhanced action potential propagation in Purkinje cell axons.

TEM revealed that axonal swellings are rich in intracellular organelles including mitochondria and endoplasmic reticulum (ER) (Fig. 3a, b yellow box inset). We also found that the majority (~80%) of axonal swellings were positive for ER-located IP3R (inositol 1,4,5-trisphosphate receptors) using immunocytochemistry (Supplementary Fig. 6), as previously reported[29]. Furthermore, the majority of axonal swelling contained close co-localization of ER and mitochondria (Fig. 3b yellow box inset), which has been linked to axonal repair in peripheral axons[30]. Surprisingly, we did not observe a significant enrichment in the density of intracellular organelles in axonal swellings compared to

control axons (Fig. 3c and Supplementary Table 1), although dense packing of organelles has been reported for axonal swellings in disease models[31], suggesting that swellings in healthy and diseased brains differ in their subcellular composition. Even without enrichment, however, all swellings contained intracellular organelles such as mitochondria and ER.

A distinguishing morphological feature of disease-related axonal swellings in human postmortem tissue is disorganized neurofilament[31]. Interestingly, we detected disorganized neurofilament in most axonal swellings from healthy young mice (Fig. 3b inset). The prevailing interpretation of disorganized neurofilament as a marker of impaired axonal transportation appears incongruous with our observation of their presence in axonal swellings that enhance axonal function, and highlight that some similarities exist between swellings in healthy brains and diseased states.

We determined that axons flanking axonal swellings were similar in diameter to control axons without swellings, suggesting that axonal swellings occur on axons that are not morphologically distinct (Supplementary Fig. 4). Axonal propagation is influenced not only by the size of the axonal diameter, but also by the thickness of the myelin sheath surrounding it. One measure that is used to understand axonal propagation is g-ratio (g-ratio = axonal diameter/(axonal diameter + total diameter including myelin sheath))[32], where g-ratios higher than a theoretical optimal value are thought to be associated with decreased propagation velocity[32]. Although we observe no changes in axosomatic delay (Figs. 1e, 2d), we find that the g-ratio for axonal swellings is higher than for control axons (Supplementary Fig. 4 and Supplementary Table 1). One explanation for this apparent discrepancy might be that g-ratio assumes a regular shape for axons[32], which is not the case for Purkinje cell axons with axonal swellings.

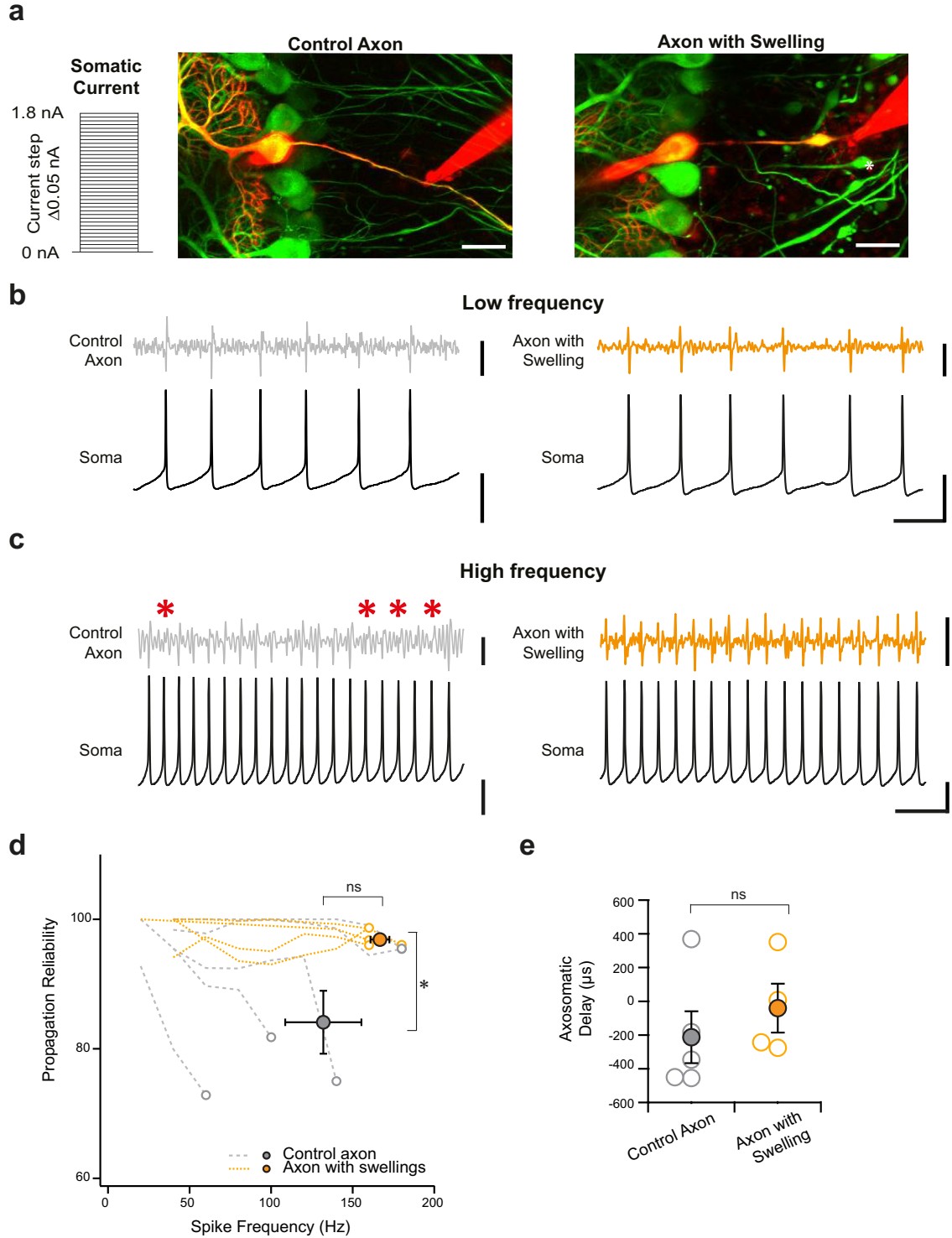

**Axonal swellings form when action potential failure rate is high in the axon**. Recent and classic studies have determined that axonal diameter can vary in response to changes in activity[33,34]. Since axonal swellings are associated with reduced axonal propagation failures, we predicted that axonal failures might be involved in their formation. To test this, we perfused a sub-saturating concentration of tetrodotoxin (10 nM; "low TTX", which is estimated to block ~75% of the $Na^+$ current in Purkinje cells)[35,36] onto live acute cerebellar slices. By recording spontaneous activity from the soma and/or axon of Purkinje cells, we observed that perfusing low TTX significantly reduced firing in axons more extensively than in Purkinje cell somas (Fig. 4a–c; $n = 17$ for somatic and axonal recordings), thereby mimicking high axonal failures pharmacologically. Through time-lapse imaging of GFP-expressing Purkinje cell axons, we discovered that swellings formed on a subset of Purkinje cell axons after 3 h of low TTX application (low TTX: $7.11 \pm 0.94$ new swellings per 100 Purkinje cells; significant difference over time, $P < 0.0001$; $n = 38$ acquisitions; Fig. 4d, e, Supplementary Table 2, and Supplementary Movie 1), while no swelling formation was observed in axons imaged over a 3 h time period without TTX (no TTX: $1.45 \pm 0.95$ new swellings per 100 Purkinje cells; $n = 6$; no difference over

**Fig. 2 Axonal swellings are associated with improved axonal propagation when Purkinje cells are driven to fire high-frequency action potentials. a** Inset left: Current was injected through the somatic patch pipette in 0.05 nA increments (Δ0.05 nA) to elicit action potential spiking at a range of frequencies to a maximum of 1.8 nA or until the cell ceased to fire action potentials. Middle and right: Representative images of recording configuration were acquired after each simultaneous dual recording. Whole-cell recordings were made from Purkinje cell somata and cells were dye-filled to visualize the axon (red Alexa 594 in pipette). Axonal recordings were made with extracellular recording electrodes from Purkinje cells without swellings (Control Axon, middle) or Purkinje cells with swellings (right). Scale bar, 20 μm. **b** With smaller current injection in the soma, Purkinje cells fired at low frequency, and axonal failures were seldom observed in both Control axons (left, gray trace) and axons with swellings (right, orange trace). Corresponding somatic recordings shown in black. (Control axon: 0.4 nA current injected; Axon with Swelling: 0.1 nA current injected). **c** With larger current injections, Purkinje cells fire at high frequencies, and axonal failures were observed more often in control axons (left, gray trace), but were less frequently observed in axons with swellings (right, orange trace). Red asterisk indicates action potential failures in the axon. Corresponding somatic recordings shown in black. (Control axon: 1.75 nA current injected; Axon with Swelling: 0.7 nA current injected). **d** Summary data shows that Purkinje cell control axons tending to show less axonal propagation reliability at higher frequencies, while axons with swellings had high axonal propagation reliability even at high frequencies. The summary data shows the average propagation reliability measurement for each individual axon at the maximal firing frequency at which they could be driven. (Control axons: average success at maximal frequency = 84.1%, maximal frequency = 128 Hz; axons with swellings: average success at maximal frequency = 96.9%, maximal frequency = 162.5 Hz; Kruskal–Wallis $H$-test: average success is significantly different, $P = 0.016$; maximal frequency is not significantly different, $P = 0.437$; control axon: $n = 5$ paired recordings; axon with swellings: $n = 4$ paired recordings. **e** The axosomatic delays between action potentials in the soma and axon are not significantly different between axons with and without swellings (two-tailed Student's $t$-test, $P = 0.45$) similar to what we observed from recordings of spontaneous action potentials (Fig. 1e). Data were presented as mean ± SEM (*$P < 0.05$; ns $P > 0.05$). Source data are provided as a Source Data file.

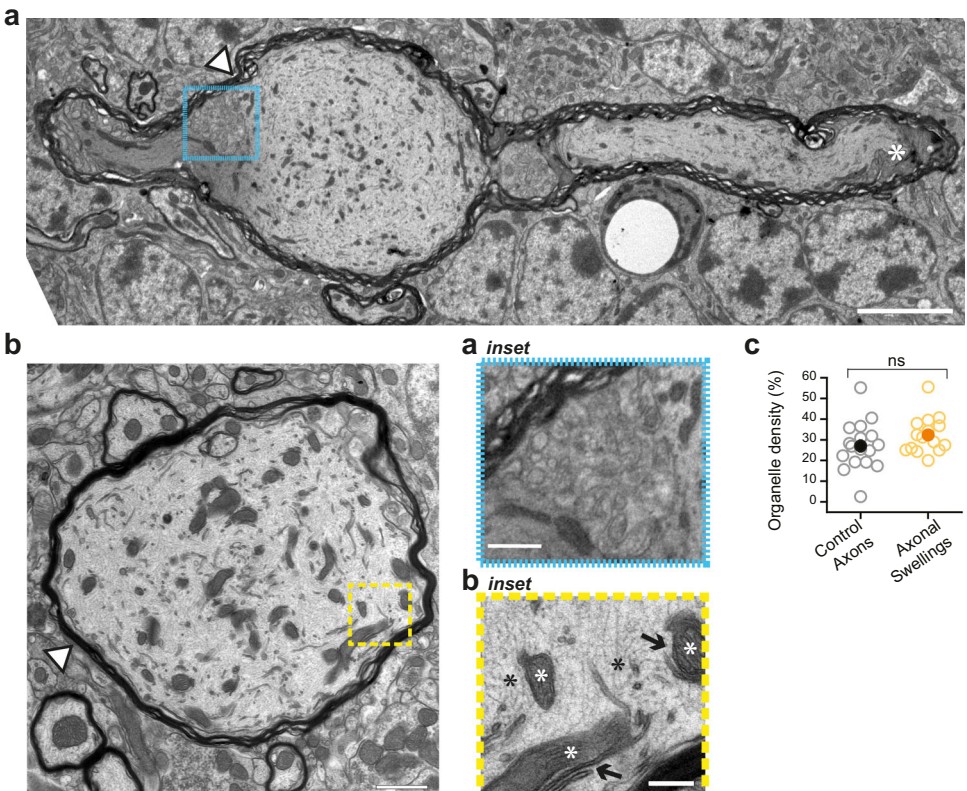

**Fig. 3 Purkinje cell axonal swellings are myelinated and are not synaptic structures. a** Representative TEM image showing longitudinal cut of axon with swelling and non-swollen sections extending from both sides of swelling (two images of longitudinal axon cut with axonal swellings were acquired). White asterisk shows perinodal protrusions (PNP) indicating proximity of node of Ranvier. Axon dips out of plane of image to the right of the swelling. Scale bar, 5 μm. Inset below middle: Region of interest delineated by cyan dashed outline in (**a**) shows periaxonal oligodendrocyte cytoplasm. Scale bar, 1 μm. **b** Representative TEM image showing crosswise cut of a putative axonal swelling (center) with five neighboring control axons, discernable by presence of dark myelin outline. Fifteen additional images of axonal swellings were acquired during TEM imaging. Scale bar, 1 μm. Inset right: Region of interest delineated by yellow dashed outline in (**b**) shows mitochondria (white asterisks) that is in close proximity to ER (black arrows), and disorganized neurofilament (black asterisks). Scale bar, 500 nm. **c** Axonal swellings are not enriched in organelles compared to control axons. (Organelles were predominantly mitochondria and ER; $n = 17$ control axons; $n = 15$ putative axonal swellings, two-tailed Student's $t$-test, $P = 0.162$). Data were presented as mean ± SEM (ns $P > 0.05$). Source data are provided as a Source Data file.

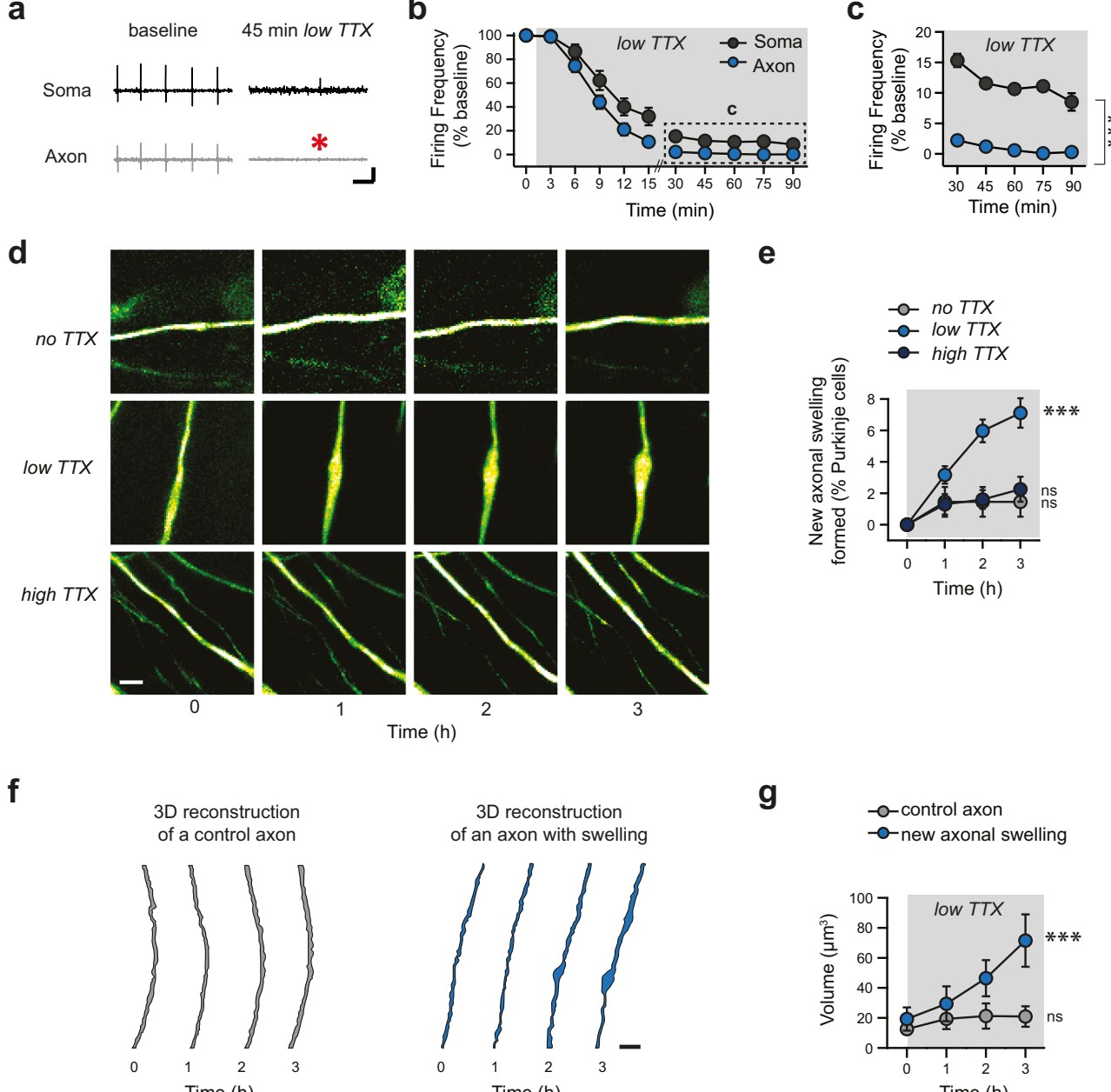

**Fig. 4 Enhancing axonal failures causes axonal swellings to form. a** Sample recordings from Purkinje cell soma (top, black) and axon (bottom, gray) prior to (left) and 45 min after (right) low TTX has been applied onto an acute cerebellar slice. Action potential amplitudes are reduced and axonal failures become prominent. Red asterisk indicates action potential failures in the axon. Scale bar, 1 mV, 25 ms. **b** Summary data showing that firing in Purkinje cell axons is reduced to a greater extent than in the soma, mimicking axonal failures. In some cases, axons and soma recordings were not from the same cell. Frequency was normalized to initial frequency prior to TTX wash-in in order to compare these cases ($n = 17$ simultaneous somatic and axonal recordings from $N = 15$ mice). **c** Replotting data in panel (**b**) showing expanded y-axis. Axonal firing rates are significantly lower than somatic firing rates, indicating that more failures occur in the axon and that this persists throughout the recording ($n = 12$ simultaneous somatic and axonal recordings from $N = 11$ mice at the last data point, as some of original 17 recordings did not last full 90 min; repeat-measured ANOVA followed by pairwise comparisons two-tailed Student's t-test using Bonferroni correction shows significant difference at each timepoint, $P < 0.001$). **d** Representative images of axons at 1-h intervals after bath application of regular ACSF (no TTX, top); low TTX (10 nM), and high TTX (200 nM). Scale bar, 5 μm. **e** Low TTX treatment results in the formation of new axonal swellings (no TTX, $n = 6$; low TTX, $n = 38$; high TTX, $n = 6$; repeated measures ANOVA showed a significant difference over time for low TTX but not the other conditions, $P < 0.001$). **f** 3D reconstruction of new axonal swelling (right) and neighboring axons without the formation of swellings (left) demonstrate that axons that later produce swellings are not morphologically distinct at the initial timepoint (time 0). Scale bar, 5 μm. **g** Summary data from 3D reconstructions reveals that volume of axons with swellings differs after 3 h, but is not different from control axons at time 0 (control axon, $n = 8$; new axonal swelling, $n = 8$; repeated measures ANOVA; $P < 0.005$). Data were presented as mean ± SEM (***$P < 0.005$; ns $P > 0.05$). Source data are provided as a Source Data file.

time, $P = 0.74$; Fig. 4d, e and Supplementary Table 2). To test whether the TTX-induced formation of focal axonal swellings was due to differential activity in the soma and axon, and not the binding of $Na^+$ channels, we applied saturating levels of TTX (200 nM, "high TTX") and found that swelling formation was minimal (high TTX: $2.26 \pm 0.79$ new swellings; $n = 6$; not significantly different over time, $P = 0.74$; Fig. 4d, e and Supplementary Table 2). This posits that the swellings formed in low TTX arise from axonal failures. To gain insight into the time window over which axonal failures are integrated, we applied low TTX briefly for 30 min followed with a firing blockade by applying high TTX for 90 min, or reversed the order, perfusing of high TTX first for 30 min followed by low TTX for 90 min. We observed that in both cases, shorter periods of axonal failure produced no formation of new swellings (Supplementary Fig. 7), suggesting that swellings only form when a sufficient number of axonal failures has occurred.

Upon pharmacologically mimicking axonal failures, we observed that axonal swellings form on a subset of axons. We wondered what differences existed between axons that did and did not form swellings. To address this, we created 3D reconstructions from time-lapse images, in low TTX conditions, of axons that formed new swellings and neighboring axons that did not (Fig. 4f). Over time, the axonal volume increased as axonal swellings formed. However, at time zero, their initial volume was indistinguishable from that of neighboring axons (control axon: $12.5 \pm 3.4$ $\mu m^3$; axon before swelling formed: $19.2 \pm 7.7$ $\mu m^3$; $n = 8$; $P = 0.96$; Fig. 4g). As expected, we observed that following 3 h of low TTX perfusion, axons that formed swellings had significantly larger volumes than those that did not (control axon: $20.9 \pm 6.8$ $\mu m^3$; axon with a newly-formed swelling: $71.4 \pm 17.4$ $\mu m^3$; $n = 8$; $P = 0.0047$; Fig. 4g and Supplementary Table 2). The similarity in volume at the initial time point suggests that there is no obvious morphological signature for axons that will form a swelling, and that swellings do not originate from the local rearrangement of axoplasm in already-thicker axons.

If axonal swellings develop in response to axonal failures, what is the signal that reports when failure occurs? Since activity is implicated, we wondered whether calcium might be playing a role. To address this, we perfused low TTX in artificial cerebrospinal fluid (ACSF) without calcium (0 mM $Ca^{2+}$), and found that the absence of calcium was sufficient to block the formation of axonal swellings ($1.86 \pm 0.58\%$; $n = 15$; not significantly different over time, $P = 0.16$, Fig. 5a, b). Meanwhile, both 2 mM (the concentration in ACSF used in all other experiments) and 3 mM of calcium resulted in robust swelling formation (2 mM $Ca^{2+}$: $5.99 \pm 1.14\%$; $n = 14$; 3 mM $Ca^{2+}$: $6.81 \pm 1.23\%$; $n = 12$; significantly different over time, $P < 0.0001$ for both, Fig. 5a, b). Importantly, 0 mM $Ca^{2+}$ in the absence of low TTX did not cause formation of axonal swellings (0 mM $Ca^{2+}$: $1.65 \pm 0.56\%$; $n = 12$; no difference over time, $P = 0.25$, Fig. 5a, b and Supplementary Table 2). If extracellular calcium is critical for axonal swelling formation, how might it be involved in signaling? Since voltage-dependent calcium channels are found in Purkinje cell axons, including T-type[37], we predicted that they might be implicated. We applied a saturating concentration of $Ni^{2+}$ (1 mM) that blocks most T-type calcium channels[37-39], but may also impact other voltage-dependent calcium channels as well[40]. We found that 1 mM $Ni^{2+}$ in the presence of low TTX prevented the formation of axonal swellings ($Ni^{2+}$ + low TTX: $1.19 \pm 0.61\%$; $n = 8$; low TTX: $6.09 \pm 1.39\%$; $n = 14$; low TTX significantly different from $Ni^{2+}$ + low TTX, Mann–Whitney $U$-test, $P = 0.0027$; Fig. 5a, c and Supplementary Table 2) while $Ni^{2+}$ without TTX had no effect (1 mM $Ni^{2+}$: $1.58 \pm 0.81\%$; $n = 6$; Fig. 5a, c). A sub-saturating concentration of $Ni^{2+}$ partially blocked the formation of axonal swellings, (Supplementary Fig. 8). These findings suggest that axonal action potential failures trigger axonal swelling via calcium entry through voltage-dependent calcium channels.

**Elevated numbers of axonal swellings are linked to enhanced cerebellar learning.** Does the increase in axonal action potential propagation fidelity associated with axonal swellings have an impact on cerebellar function? To address potential functional changes, we assayed mice with Rotarod (Fig. 6a), a motor task implicated in motor coordination and learning[19], and took advantage of the natural variability in learning that is observed across young adult mice (Fig. 6b). After performing behavioral assays, we quantified the number of Purkinje cell axonal swellings found in the granule cell layer in lobule III of the vermis, a cerebellar region that is important for locomotion[41]. We observed variability in the number of swellings across animals (Fig. 6b) which positively correlated with the amount of learning on the Rotarod task ($R = 0.544$; $P = 0.011$; Fig. 6c). High-learning mice had significantly more axonal swellings (Fig. 6d) than low-learning mice (low learner: $29.8 \pm 3.3\%$ axonal swellings; $n = 10$; high learners: $39.2 \pm 3.0\%$ axonal swellings; $n = 11$; $P = 0.046$; Fig. 6e). These results highlight the positive effect that axonal swellings have on cerebellar-related motor learning. Based on these data, we developed a Monte Carlo simulation to understand the differences in information content in a network of Purkinje cells with varying numbers of axonal swellings. This model enabled us to estimate the amount of learning that can be accounted for by variation in axonal swelling occurrence, and relate this to the amount of information that is lost due to axonal failures in this network. The amount of learning that the model predicted varied dramatically depending on the number of swellings. Since this model was constructed based on our electrophysiological and behavioral data, it allows us to link how much variability in learning we might expect from a network that contains different numbers of Purkinje cells with axonal swellings. This simulation supports our experimental findings that across-animal differences in the number of axonal swellings can account for some of the variability we observe in learning across animals due to information loss along axons in axons without swellings (Supplementary Fig. 9).

We further investigated the difference in motor learning with a cerebellar-specific assay, the Erasmus ladder[42]. We trained mice to cross an Erasmus ladder every day for 4 days (Fig. 6f), noting the amount of learning that occurred. As we observed for Rotarod, there was significant heterogeneity in learning across mice (Fig. 6g). Still, learning was positively correlated with the number of axonal swellings found in lobule III of the cerebellar vermis ($R = 0.776$; $P = 0.008$; Fig. 6h, i). When mice were grouped into low and high learners on the Erasmus ladder, we found that high learners had significantly more axonal swellings than low learners (low learners: $34.9 \pm 4.2\%$; $n = 5$; high learners: $46.6 \pm 2.2\%$; $n = 5$; $P = 0.039$; Fig. 6j).

To determine whether axonal swellings influence behavior in a cerebellar region with a well-defined associated behavior, we tested the adaptation of the vestibular ocular reflex (VOR; Fig. 6k). VOR adaptation is known to be encoded by Purkinje cells in the flocculus[43], which differ in intrinsic firing rate from those in lobule III[44,45]. There is relatively little variability across animals for VOR adaptation (Fig. 6l). These experiments were also conducted in C57Bl/6 J mice and axonal swellings were labeled with IP3R, which we have shown labels the majority (~80%), but not all, of axonal swellings in both Lobule III and the flocculus (Supplementary Fig. 3). We observed a positive correlation between the number of axonal swellings and learning in the flocculus that was reminiscent of the correlation seen in the

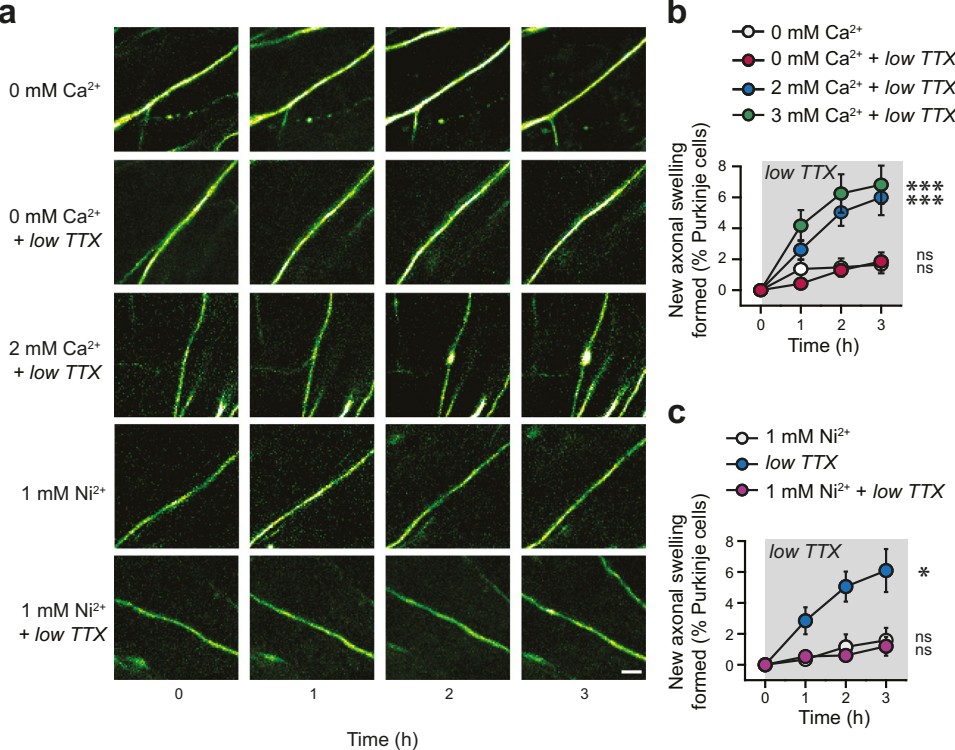

**Fig. 5 Axonal swelling formation requires calcium influx through voltage-gated Ca$^{2+}$ channels. a** Representative time-lapse images during bath application of ACSF containing no Ca$^{2+}$ (0 mM Ca$^{2+}$, top), 0 mM Ca$^{2+}$ with low TTX (second row), low TTX with 2 mM Ca$^{2+}$ (third row). The formation of axonal swellings in low TTX is occluded by the presence of 1 mM Ni$^{2+}$ (fourth row), and Ni$^{2+}$ without TTX does not produce axonal swelling formation. Scale bar, 5 μm. **b** Axonal swellings do not form in *low* TTX with 0 mM Ca$^{2+}$, but do form in both 2 mM and 3 mM Ca$^{2+}$ (repeated measures ANOVA; 0 mM Ca$^{2+}$, $n = 12$, $P = 0.25$; 0 Ca$^{2+}$ + low TTX, $n = 15$, $P = 0.16$; 2 mM Ca$^{2+}$ (2 mM) + low TTX, $n = 14$, $P < 0.001$; 3 mM Ca$^{2+}$ + low TTX, $n = 12$, $P < 0.001$). **c** Summary data showing low TTX ACSF containing 1 mM Ni$^{2+}$ occludes the formation of axonal swellings. (repeated measures ANOVA; ACSF containing 1 mM Ni$^{2+}$, $n = 6$, $P = 0.78$; low TTX, $n = 14$, $P < 0.001$; 1 mM Ni$^{2+}$ + low TTX, $n = 8$, $P = 0.80$). Data were presented as mean ± SEM (***$P < 0.001$; *$P < 0.05$. ns $P > 0.05$). Source data are provided as a Source Data file.

anterior vermis, although not significant ($R = 0.627$; $P = 0.052$; Fig. 6m, n); furthermore, no significant difference was observed between the low and high learners (low learners: 13.2 ± 1.5%; $n = 5$; high learners: 17.8 ± 2.2%; $n = 5$; $P = 0.12$; Fig. 6o). This suggests that variability in axonal swellings in the flocculus does not account for the variability in learning in this task. However, taken together, our data suggest that axonal swellings have a modest but positive impact on cerebellar function, a trend that is consistent with their ability to ameliorate axonal spike fidelity.

As we observed fewer axonal swellings in the flocculus than in the anterior vermis of mice, we harnessed tissue clearing and light-sheet imaging techniques to determine whether the number of swellings varied across cerebellar region (Supplementary Movie 2). Focusing on the cerebellar vermis, we found that the number of swellings varied dramatically across cerebellar lobules, with lower numbers of swellings in posterior lobules (Supplementary Fig. 10). Having conducted the Rotarod task prior to imaging, we were also able to observe the positive correlation between the density of axonal swellings and learning: greater density of swellings tended to coincide with more learning (Supplementary Fig. 10).

## Discussion
Here we describe that Purkinje cell axonal swellings in young mice are associated with enhanced axonal fidelity and cerebellar performance. Using targeted paired recordings, we report that Purkinje cell axons presenting focal swellings have significantly fewer axonal failures than Purkinje cell axons without swellings, with no detectable changes in firing properties. This is observed both for spontaneous firing rates and when Purkinje cells are driven to fire at higher frequencies. Purkinje cell axonal swellings are myelinated, show no evidence of being synaptic structures, do not show accumulation of intracellular organelles, but are frequently observed in proximity to nodes of Ranvier or at locations enriched with oligodendrocytic cytoplasm. We wondered if axonal failures are instrumental in the formation of axonal swellings, and found that by mimicking high axonal failures, we could induce the formation of swellings on Purkinje cell axons within 2 to 3 h. Axonal swelling formation required extracellular calcium entry and was blocked by a voltage-dependent Ca$^{2+}$ channel blocker, suggesting that neurons detect axonal failures by the integration of calcium influx through voltage-dependent channels. Finally, using three different cerebellar-related behavioral assays, we demonstrated a positive correlation between motor learning and the number of axonal swellings in related cerebellar structures in young adult mice. Light-sheet imaging revealed that the density of axonal swellings even varies within an animal across cerebellar lobules. These data suggest that there appears to be a behavioral read-out to the enhancement of axonal propagation associated with axonal swellings.

Modeling studies have mostly predicted that action potentials moving across axons that swell will be delayed, filtered, or fail to propagate[12–14], although this depends greatly on the precise geometry of the swelling, with small differences leading to large and at times opposite outcomes[14]. Furthermore, these models have typically focused on non-myelinated axons, so it is unclear how they apply to myelinated axons like those of Purkinje cells.

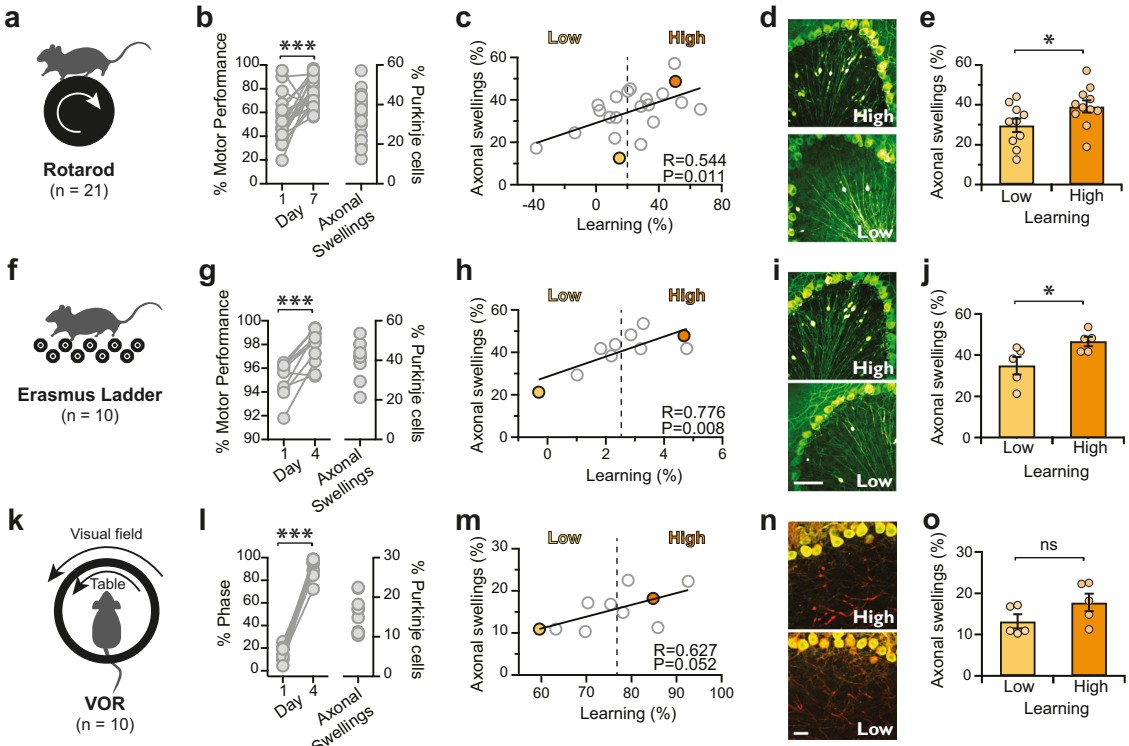

**Fig. 6 Axonal swelling density is positively correlated with cerebellar-related learning. a** Rotarod assay. **b** Variability in motor learning (left; two-tailed paired Student's $t$-test, $P < 0.001$) and the density of axonal swellings in anterior vermis lobule III (right) across mice. **c** The variability in motor learning is positively correlated with the density of axonal swellings (two-tailed Pearson correlation, $R = 0.544$, $P = 0.011$). The filled orange marker indicates representative high learner in (**d**), while the yellow marker indicates the low learner in (**d**). **d** Representative images of anterior cerebellar vermis from high learners (top) and low learners (bottom). Note the differences in the number of axonal swellings. **e** When mice are separated into high and low learner groups (Low learners, $N = 10$ animals; High learners, $N = 11$ animals; two-tailed Student's $t$-test, $P = 0.046$), they have significantly different numbers of axonal swelling, with high learners having more than low learners. **f** Erasmus ladder assay. **g** Variability in motor learning (left; two-tailed paired Student $t$-test, $P = 0.001$) and anterior lobule swelling density (right) are also observed with Erasmus ladder. **h** This variability is positively correlated across animals (two-tailed Pearson's correlation, $R = 0.776$, $P = 0.008$). **i** Representative images from high (top) and low (bottom) learners on the Erasmus ladder assay. **j** High learners on Erasmus ladder show more axonal swellings in lobule III of the vermis than low learners (Low learners, $N = 5$ animals; High learners, $N = 5$ animals; two-tailed Student's $t$-test, $P = 0.039$). **k** VOR assay. **l** Some but less variability in learning is observed in VOR (left; two-tailed paired Student's $t$-test, $P < 0.001$) and in the density of axonal swellings in the flocculus (right). **m** Variability in learning and axonal swelling density in the flocculus is correlated across animals for VOR (two-tailed Pearson's correlation, $R = 0.627$, $P = 0.052$). **n** Representative images showing axonal swellings (stained with IP3R) in high learners (top) and low learners (bottom) for VOR task. **o** The density of axonal swellings in the flocculus is not significantly different in low and high VOR learners (Low learners, $N = 5$ animals; High learners, $N = 5$ animals; two-tailed Student's $t$-test, $P = 0.122$), perhaps because only a subset of swellings are labeled with IP3R, although the same trend is observed as for the Rotarod and Erasmus ladder tasks. Data were presented as mean $\pm$ SEM (***$P < 0.005$, *$P < 0.05$; ns $P > 0.05$). Source data are provided as a Source Data file.

For myelinated axons, g-ratio relates the thickness of myelination to an axon's diameter; and there is a theoretical optimal g-ratio that produces the fastest propagation velocity in an axon[32]. However, g-ratios assume regular axonal shape[32] and thus are likely not an ideal measurement for axonal swellings, where diameter varies dramatically. Understanding how a local axonal diameter increase leads to improved axonal spike fidelity is important. One potential explanation is suggested by recent findings from Cohen and colleagues, who report that perinodal structures play a role in saltatory conduction[28]. Given that axonal swellings are located proximal to perinodal structures (Supplementary Fig. 5), their location along the axon may contribute to their impact on propagation fidelity.

Axonal plasticity enables neurons to modulate their excitability and optimize neuronal output. For example, activity-dependent modulation of the AIS, where action potentials are initiated, enables neurons to homeostatically adapt to alterations in their activity[2,46–48]. The formation of axonal swellings is reminiscent of homeostatic synaptic and intrinsic alterations like synaptic scaling or firing rate homeostasis that optimize network output[49,50]. In

this case, however, axon propagation fidelity appears to be the output that is being optimized. Activity-dependent myelination is another form of plasticity that has been observed in several brain regions, where axons with heightened activity trigger oligodendrocytes to increase myelination[51–53]. Since Purkinje cells with axonal swellings fire at similar frequencies to those without (Fig. 1f), the plasticity associated with axonal swellings appears distinct from mechanisms of activity-dependent myelination.

Understanding how neurons decode axonal failures leading to the formation of axonal swellings is an important question to be answered. The signaling pathway involving calcium influx may take place directly in the axon, since voltage-dependent calcium channels are expressed in axons[37]. However, it is also possible that signals occur in distal locations that are then relayed to the axon, or that signaling originate from the surrounding myelin sheath rather than from the axon itself[54], perhaps explaining the relatively slow time course of swelling formation.

We observed a correlation between the density of axonal swellings and different cerebellar-related forms of behavioral learning. These findings were surprising to us, since learning and

memory are thought to be largely determined by synaptic plasticity[55]. Yet many factors contribute to successful learning, including the reliable transmission of information. Indeed, we found that the number of axonal swellings correlates with motor performance on the last day of the motor assay to a similar extent as with the amount of learning (Supplementary Fig. 11). A parsimonious explanation of our data is that the correlation between learning and axonal swellings arises because information becomes more reliably propagated in the cerebellum, and that the effect on learning could thus be indirect. Chaisanguanthum and colleagues recently showed that even a single extra action potential in Purkinje cells in the cerebellar flocculus could impact eye-movement behavior variability[56]. These data argue that reliable transmission of action potentials in axons is essential for the engagement of appropriate forms of synaptic plasticity in the brain.

Purkinje cells transmit information in vivo with two distinct patterns: simple spikes and complex spikes. Purkinje cell axons have been reported to have relatively low propagation failure rates for simple spikes with higher failures for complex spikes[23,35,37]. Our study has focused on spontaneous action potentials, or when action potentials are driven at higher frequencies with current injections, which correlate with simple spikes in vivo. How and if axonal swellings impact complex spike propagation down the axon is an intriguing question to be explored.

Purkinje cell axons are important because they transmit information from the cerebellar cortex to downstream targets. Interestingly, similar axonal swellings, which are sometimes called spheroids, have been observed in several locations, including frontal lobe white matter[57,58], hippocampus[59], basal ganglia[60], brain stem[61], and spinal cord[62,63]. Axonal swellings in other brain regions have frequently been associated with neurodegenerative diseases[57–60,62], although axonal swellings have also been reported in other brain regions from healthy humans[61,63] and animal models[63]. Whether axonal swellings on axons in other brain regions are associated with enhanced action potential propagation as we have observed, or whether our results arise from a unique property of Purkinje cell axons, remains to be determined.

Axonal swellings in neurodegenerative diseases have typically been regarded as morphological signatures of neuronal dysfunction[8–11]. Although one suggestion raised by our findings is that disease-related axonal swellings might also serve an adaptive role, several alternative explanations exist. It is possible that disease-related axonal swellings differ from those observed in healthy states, as mice used in our study were young and healthy and did not suffer from a neurodegenerative disease. Furthermore, axonal swellings in neurodegenerative diseases exhibit heterogeneity in their myelination; that is, some diseases are associated with axonal swellings that are myelinated while others are associated with predominantly unmyelinated swellings[64]. Differences in underlying axonal dysfunction may produce axonal swellings that look similar at the light-microscopic level, but that are nonetheless structurally and functionally distinct. It would be interesting to test the functional properties of axonal swellings arising from disease models to gain insight into whether disease-related swellings show similar properties to those characterized here. Regardless of whether they are similar or distinct from one another, our findings highlight the importance of empirically determining the impact of morphological specializations on neuronal function.

In conclusion, our data suggest that Purkinje cell axonal swellings are an adaptive morphological feature that enables an axon to maintain optimal axonal propagation. In light of the central role that Purkinje cell axonal conductance plays in transmitting information from the cerebellar cortex, our findings

suggest that axonal swellings likely represent a homeostatic form of plasticity that contributes to the optimization of cerebellar function[50].

## Methods

**Animals.** We used male and female *pcp2-tau-eGFP* mice[16,17] to characterize the functional properties of axonal swellings as well as for time-lapse visualization of their formation and for Rotarod and Erasmus Ladder behavioral studies. C57BL/6 J mice were used for VOR behavioral study. All animal procedures were approved either by the McGill Animal Care Committee, in accordance with guidelines established by the Canadian Council on Animal Care, or for experiments from Fig. 5k–o by the Dutch Ethical Committee for animal experiments in accordance with the Institutional Animal Care and Use committee at the Erasmus Medical Centre. Mice were housed in rooms with 12/12 light/dark cycle, at 20–24 °C, with 40–60% humidity.

**Behavior.** Male and female mice were used at 1–2 months of age for all behavioral studies.

*Rotarod assay.* We used a Rotarod (Stoelting Europe, Dublin, Ireland) as previously described[19] to assess the natural variability of individual mice motor coordination. After an hour of acclimatization, mice were placed on an accelerating (from 4 to 40 RPM, over 5 min) Rotarod and latency to fall (4 trials/day for 7 days) was recorded. Motor learning was determined by subtracting the average time on the rod of the last two trials for day 1 from day 7.

*Erasmus ladder assay.* We used the Erasmus Ladder (Noldus Inc., Wageningen, GE, Netherlands) to assess motor learning[42]. Mice walked across a horizontal ladder (42 trials/day for 4 days), which consisted of two parallel rows of 37 pressure monitored rungs between two dark chambers. Mice typically used short steps (stepping between two upper rungs) to traverse the ladder. The change in the number of short steps across days was used to measure motor learning in individual animals.

*VOR assay.* Mice were equipped with a construct for immobilization ("pedestal") under general anesthesia with isoflurane/O2. After a 2–3 days of recovery, mice were head-fixed with the body loosely restrained in a custom-made restrainer and placed in the center of a turntable (diameter: 60 cm) in the experimental set-up. A round screen (diameter 63 cm) with a random dotted pattern ("drum") surrounded the mouse during the experiment. VOR Phase reversal was induced by training mice over 4 days (6 × 5 min per day) using in-phase sinusoidal drum and table rotation at 0.6 Hz (amplitude table 5° on all days, drum 5° on day 1, 7.5° on day 2, 10° on day 3–4) and probed by recording VOR in the dark before and after training sessions. A CCD camera fixed to the turntable monitored the eyes of the mice using eye-tracking software (ETL-200, ISCAN systems, Burlington, NA, USA). For eye illumination during the experiments, two infrared emitters (output 600 mW, dispersion angle 7°, peak wavelength 880 nm) were fixed to the table and a third emitter, which produced the tracked corneal reflection, was mounted to the camera and aligned horizontally with the optical axis of the camera. Eye movements were calibrated by moving the camera left-right (peak-to-peak 20°) during periods that the eye did not move[45]. Gain and phase values of eye movements were calculated using custom-made Matlab routines (MathWorks, Natick, MA, USA).

**Acute slice preparation.** Acute slices were prepared from male and female young juvenile mice (postnatal (P)9–16), when axonal swellings are numerous[4]. Mice were deeply anesthetized using isoflurane and checked for foot-paw reflex. Mice were decapitated, and the brain quickly removed in ice-cold ACSF (in mM: 125 NaCl, 2.5 KCl, 2 CaCl₂, 1 MgCl₂, 1.25 NaH₂PO₄, 26 NaHCO₃, and 20 glucose, bubbled with 95% O₂ and 5% CO₂ to maintain pH at 7.3; Osmolarity 320 ± 5 mOsm). Parasagittal cerebellar vermis slices (200 μm thick) were made using a Leica VT 1000 S vibratome. Slices were incubated in ACSF at 37 °C for 45 min and then incubated at room temperature (RT) until used for experiments. All chemicals were purchased from Sigma-Aldrich (Oakville, ON, Canada), unless otherwise specified.

**Electrophysiology.** All recordings were made from Purkinje cells in lobule III from the cerebellar vermis, using a Multiclamp 700B amplifier (Molecular Devices, Sunnyvale, CA, USA), at 33 °C. An upright microscope (Scientifica, Uckfield, UK) combined with a custom-built two-photon Ti:Sapphire laser (MaiTai; Spectra Physics, Santa Clara, CA, USA) was used to identify Purkinje cells soma with intact axons (GFP). Borosilicate patch pipettes were pulled using a P-1000 puller (Sutter Instruments, Novato, CA, USA). For loose cell-attached or extracellular recording of Purkinje cells soma and axons, glass pipettes were dipped in CdSeS/ZnS alloyed quantum dots (Sigma-Aldrich, Oakville ON, CA), filled with ACSF, and were visually positioned using the scanning two-photon laser as previously described[15]. For whole-cell current-clamp experiments, patch pipettes (2−7 MΩ) targeting Purkinje cell soma were filled with an internal solution containing (in mM): 122 K-gluconate, 3 KCl, 10 K-HEPES, 2 MgCl₂, 2 MgATP, 0.4 NaGTP, 0.05 EGTA, and

10 phosphocreatine-(di)tris, adjusted with KOH to pH 7.2–7.4, and with sucrose to 290–295 mOsm[17]. AlexaFluor 594 (50 mM, ThermoFisher, Burlington, ON, CA) was added to the internal solution to visualize Purkinje cell axons. In current-clamp experiments, we targeted filled Purkinje cell axons and performed extracellular recordings using glass pipettes (~1 MΩ) filled with ACSF that were dipped in either quantum dots or filled with AlexaFluor 594. All electrophysiological recording of Purkinje cells soma and axons were performed using a custom-designed Igor Pro 6.37 acquisition software (Wavemetrics, Portland, OR, USA). Axons were recorded in the granule cell layer, downstream from axonal swellings (50–200 μm from the parent soma), or an equivalent distance down a control axon without swellings. Approximately 30% of axons display swellings at this age[4]. We only recorded from axons with single swellings, which were the majority of axons with swellings (98.7% or 950/963 of axons with swellings had single swellings, $N = 4$ animals; Supplementary Fig. 1). In extracellular experiments (Fig. 1), Purkinje cell somatic and axonal action potentials were passively recorded. In current-clamp experiments (Fig. 2), Purkinje cells were held at −60 mV and were injected with 500 ms-long current steps with 10 s intervals. Current injections started with an amplitude of 0.05 nA and increasing by 0.05 nA up to 1.8 nA or until the cell ceased to fire action potentials.

**Imaging.** Parasagittal cerebellar slices were imaged with a custom-built two-photon microscope (Scientifica) with a Ti:Sapphire laser (MaiTai; Spectra Physics, Santa Clara. CA, USA) tuned to 890 nm (GFP) or 775 nm (non-GFP). Images acquisition was done using ScanImage 3.7 running in MatLab 2011B (Mathworks, Natick, MA, USA). For live imaging, cerebellar slices were kept alive by continuously perfusing buffered ACSF (with drugs) at 33 °C. For fixed tissue, slices from the vermis were imaged on a LSM800 laser scanning confocal microscope (Zeiss, Oberkochen, Germany), while slices from the flocculus were imaged using an LSM700 laser scanning confocal microscope (Zeiss).

**Pharmacology.** Tetrodotoxin (TTX; Biotium Inc., Fremont, CA, USA) was used at a concentration of 10 nM (low TTX) or 200 nM (high TTX) and Nickel chloride (NiCl; Sigma-Aldrich, Oakville, ON, Canada) was used at a concentration of 1 mM in ACSF. Extracellular $Ca^{2+}$ was also manipulated by changing its concentration in the ACSF (3 or 0 mM) while keeping the concentration of other positively-charged divalent ions (e.g., $Mg^{2+}$) constant.

**Immunocytochemistry.** Mice were anesthetized using an intraperitoneal (IP) injection of Avertin (2,2,2-tribromoethanol; dosage: 0.25 ml/10 g body weight), and transcardially perfused with 4% paraformaldehyde (PFA; Electron Microscopy Sciences, Hatfield, PA, USA). Perfused brains were removed and stored at 4 °C on a rotary shaker at 70 RPM for 24 h in 4% PFA. Brains were then transferred into phosphate-buffered saline (PBS) with 0.05% sodium azide. Brains were sliced on a Leica vibratome 3000 into 100-μm-thick parasagittal slices or on a cryostat Leica CM into 40-μm-thick coronal slices. In some cases, brain slices were briefly washed in a 0.1 mg/ml pepsin solution dissolve in 1x phosphate buffer (PB), for 5 min at RT for antigen retrieval. Slices were washed in 1x PBS—0.4% Triton X, blocked with 5% BSA, and primary antibodies (mouse anti-CASPR, used at a dilution of 1:200, Antibodies Incorporated, Davis, CA, US, Cat#75-001; rabbit anti-IP3R, used at a dilution of 1:200, Abcam, Cambridge, UK, abID#5804) were applied to slices for 3 days while incubated on a rotary shaker at 70 RPM at RT. After washing, secondary donkey anti-mouse or anti-rabbit antibodies conjugated to Alexa Fluor-594 (1:500, Life Technologies, Carlsbad, CA, USA, product # A12203; 1:500, Jackson Immunoresearch Labs, West Grove, PA, USA, product # 711-585-152, respectively) were applied to slices while incubated on the rotary shaker for 90 min at RT. Slice were mounted on slides with Prolong gold anti-fade mounting media (Life Technologies). For imaging of axonal swellings in the flocculus, experiments differed from above in the following manner: mice were deeply anesthetized through IP administration of sodium pentobarbital, and brains were postfixed for 1 h instead of 24 h, and were subsequently transferred to a 10% sucrose solution overnight at 4 °C. The following day they were embedded with 10% sucrose/14% gelatin (Wako) and placed in a 30% sucrose/10% formaldehyde for 1 h at RT, and then were switched to a 30% sucrose solution overnight at 4 °C. Embedded brains were sectioned transversally into 40-μm-thick slices with a freezing microtome. Sections were rinsed with 0.1 M PB and incubated for 2 h in 10 mM sodium citrate (pH 6) at 80 °C for 2 h for antigen retrieval. About 10% horse serum was used instead of 5% BSA to block nonspecific binding and antibodies were applied for 2 instead of 3 days. Primary antibodies used were Calbindin (1:7000 mouse monoclonal, Sigma C9848) and IP3R (1:1000 rabbit polyclonal, Abcam 5804). Secondary antibodies used were donkey anti-rabbit coupled to Alexa Fluor-488 (1:500, Life Technologies, product # A32790) or goat anti-mouse Cy3 (1:200; Life Technologies, product # A10521). Sections were mounted on coverslips in Chromaluin (genatin/chromate) and covered with Mowiol mounting medium (Polysciences Inc., Bergstrasse, Germany).

**Electron microscopy.** Seven P31–33 male and female mice were anesthetized using an IP injection of Avertin and transcardially perfused with a combination of 2% glutaraldehyde (Electron Microscopy Sciences) and 2% paraformaldehyde (Electron Microscopy Sciences) in 1X PBS at a perfusion rate of 5 mL/min. The brain

was then removed and placed in 2.5% glutaraldehyde in 0.1 M sodium cacodylate (Electron Microscopy Sciences) buffer at 4 °C, overnight. Brains were dissected to isolate the anterior lobules of the cerebellar cortex. TEM was performed of axons located in the anterior cerebellar lobules 3, 4, and 6. Lobules were kept at 4 °C in fixative no longer than a week before being processed for TEM. Subsequently, samples were washed three times with 0.1 M sodium cacodylate washing buffer (Electron Microscopy Sciences) for 1 h. The samples were then postfixed using 1% aqueous osmium tetraoxide (Mecalab, Montreal, QC, Canada) and 1.5% aqueous potassium ferrocyanide for 2 h, followed by three washes of washing buffer for a total of 15 min. Then, samples were dehydrated with acetone (Fisher Scientific) in increasing concentrations (30, 50, 70, 80, 90, and 3 × 100%) for 8–15 min per concentration. Samples were then infiltrated with Epon (Mecalab) in acetone as follows: 1:1 overnight, 2:1 the next day, 3:1 the following night, and pure Epon the last day for 4 h. Samples were embedded and allowed to polymerize in a 60 °C oven for 48 h. Samples were trimmed and cut at 90–100 nm thick sections with UltraCut E ultramicrotome (Leica Microsystems, Wetzlar, HE, Germany, formerly Reichert-Jung) and placed onto slotted grids (Electron Microscopy Sciences). Finally, sections were stained with uranyl acetate (Electron Microscopy Sciences) for 8 min, followed by Reynold's lead (Electron Microscopy Sciences) for 5 min. Samples were imaged using FEI Tecnai G2 Spirit Twin Cryo-TEM (FEI Company, Hillsboro, OR, USA) at 120 kV and visualized with an AMT XR80C 8 megapixel CCD camera (Advanced Microscopy Techniques, Woburn, MA, USA).

**Tissue clearing.** Mice were anesthetized and transcardially perfused as above. The brain was removed and transferred into 4% PFA and stored at 4 °C on a rotary shaker at 70RPM for 24 h after which the cerebellar cortex was isolated and transferred into a hydrogel solution (10% v/v 10X PBS, 10% v/v 40% Acrylamide; 0.025% v/v 10% VA-044, and filled up to 15 ml with $ddH_2O$) at 4 °C for 24 h. The cerebella were then transferred to the X-Clarity Polymerization system (Logos Biosystem, Annandale, VA, USA) at 37 °C, under 90 kPa vacuum for 3 h, followed by the X-Clarity Tissue Clearing system, using the electrophoretic tissue clearing solution (Logos Biosystem, Annandale, VA, USA) set at a current of 1 Amp at 37 °C for 24 h. A Lightsheet Z.1 with the Zen 2014 SP1 (black edition) software (Zeiss, Oberkochen, Germany) was used to image the whole cerebellum.

**Modeling.** A Monte Carlo simulation was done in Python 3.6 with a Tkinter graphical user interface. A simulation of 40 Purkinje cells was run 5000 times using experimentally-determined axonal failure rates (Fig. 1c) and action potential firing rates (Fig. 1f) to populate the simulation. Each 200,000-cell network of Purkinje cells was modeled varying the percentage of Purkinje cells with axonal swellings in the network: 0, 25, 50, 75 and 100%. The amount of learning for each network was then projected based on experimental data (Fig. 6c), and was run 100 times for each network containing 0, 25, 50, 75, and 100% of Purkinje cells with swellings.

**Data analysis.** We used a custom-designed Igor Pro 8 software for electrophysiology data analysis of spike timing, frequency, regularity, and failures. Current injection data was binned into 20 Hz frequency bins for Fig. 2. Image analysis of two-photon image stacks was conducted out with Fiji/ImageJ2 version 2.0.0-rc-69/1.52p or Zen 2014 SP1 (black edition: Zeiss). Lightsheet image processing was done using Zen 2.5 (blue edition), Imaris File Converter and Stiching 9.2.1 software, and analysis was completed using Imaris 9.3.0 software (Bitplane Inc., Zurich, Switzerland). For axonal reconstructions in Fig. 3, we used Neurolucida 10 software (MBF Biosciences, Williston, VT, USA) to reconstruct an axonal swelling and the sections of axon flanking both sides of the swelling, in addition to an equivalent length of neighboring axon without a swelling. For measurement of velocity, axons were reconstructed from the recording location to the soma using Simple Neurite Traces, a plugin for ImageJ. For EM images, axons were unbiasedly identified and imaged using the presence of myelin. Using Fiji/ImageJ2, the axonal circumference was traced and minimum Feret diameter was measured. In one case where minimum Feret diameter was not applicable (i.e., in Fig. 3a when the axon was not spherical but rather was observed in cross-section, and thus rectangular in shape), the diameter was taken by averaging 4 diameters along the axon. Myelin thickness was quantified by measuring myelin width at four cartesian points, although care was taken to avoid locations where myelin infiltration occurred. Organelle density was measured by tracing organelle bodies and was quantified as the ratio between the area covered by organelles to the total axonal area. Organelles included mitochondria, ER, and putative endosomes and lysosomes, and excluded neurofilament.

**Statistics.** Data normality and equality of variance were determined using Shapiro–Wilk and Levene's test. Multiple comparisons with repeated measures ANOVA were performed using IBM SPSS Statistics 27 (IBM Corp., Armonk, NY, USA). All reported data are interaction effect of conditions over time corrected for sphericity using Greenhouse–Geisser test, with post hoc analysis with Bonferroni correction were performed between all conditions over time. Due to low degree of freedom, the assumption of normality is not always respected. To address that concern, nonparametric multiple comparisons were made using Kruskal–Wallis H-test to compare all conditions at the last time point, followed by Mann–Whitney U-test corrected with Bonferroni. Simple comparisons were made using either paired

or unpaired two-tailed Student's *t*-tests for parametric data, or the Mann–Whitney *U*-test for nonparametric data, and correlations were made using the Pearson (*r*) correlation test in SPSS software. Data were reported as mean ± SEM. Statistical comparisons were made with the level of significance (α) set at $P* < 0.05$; $P** < 0.01$; $P*** < 0.005$, unless corrected with Bonferroni.

**Reporting Summary**. Further information on research design is available in the Nature Research Reporting Summary linked to this article.

## Data availability
All relevant data are available from the corresponding author upon reasonable request. Source data for each figure is provided as a Source Data file. Source data are provided with this paper.

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

## Acknowledgements

We thank S. du Lac and M. Häusser for generously providing the mice, which originated in the du Lac lab. We thank J. Sjöström for custom software for action potential failure analysis, E. Ruthazer, R. A. McKinney, J. Sjöström, and A. Fournier for thoughtful input on the project, and comments from D. Jaarsma. We thank members of the Watt lab (past and present) for input, support, and feedback on the project and the manuscript as well as N. Recio and A. Huang for technical contribution. We thank Claire Brown and other members of the McGill University Life Sciences Complex Advanced BioImaging Facility (ABIF) and Jackie Vogel and other members of the McGill University Integrated Quantitative Biology Initiative (IQBI) for technical resources and support with X-clarity, lightsheet, and confocal imaging and image analysis. We thank the members of the McGill University Facility for Electron Microscopy Research (FEMR) for technical assistance with TEM. We thank members of the McGill GCC animal care facility for technical assistance, and L. Post and S. Sahin for technical contributions to the VOR data. This work was supported by a Postgraduate PGS-D Scholarship-Doctoral (D.L.-O.) and Summer Undergraduate Research Awards (C.H.L. and C.R.) from the National Science and Engineering Research Council of Canada (NSERC), a McGill Integrated Program in Neuroscience (IPN) Returning Student Award (C.A.S.), and a New Investigator (Nou-veau Chercheur) Grant from the Fonds de Recherche Nature et Technologies de Quebec (A.J.W., 189153), a European Research Council grant (M.S., ERC-Stg No. 680235), a Canadian Institutes of Health Research (CIHR) Operating Grant (A.J.W., 130570) and Project Grant (A.J.W., 153150), an NSERC Discovery Grant (A.J.W., 05118), and a Canadian Foundation for Innovation (CFI) Leaders Opportunity Fund (A.J.W., 29127).

## Author contributions

D. L.-O. designed and ran experiments and analyzed data for all Figures, and helped write the manuscript. K.M.G. and A.S.-D. designed and ran experiments and analyzed data for Fig. 2 and helped write the manuscript. F.G.C.B. designed and ran experiments for Fig. 6 and helped write the manuscript. C.A.S. designed, ran experiments, and analyzed data for Fig. 3 and helped write the manuscript. P.d.V.d.B. ran experiments for Fig. 5. C.H.L. designed and ran experiments for Fig. 4 and Fig. S10 and helped write the manuscript. C.V.E. designed and ran Monte Carlo simulation for Fig. S9. C.R. ran experiments for Fig. 1. P.L.F. helped interpret data for Fig. 3 and helped write the manuscript. M.S. designed experiments and analyzed data for Fig. 6 and helped write the manuscript. A.J.W. conceived of the project, designed experiments and analyzed data, supervised the project, and wrote the manuscript.

## Competing interests

The authors declare no competing interests.

## Additional information

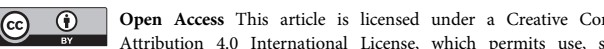

