## [Peer Review File · Nature Communications]

Reviewer #1 (Remarks to the Author):

The manuscript by Lang-Oulette and colleagues presents a very exciting set of results. The so-called torpedoes in Purkinje cell axons have been topic of investigation in the context of pathology for several decades but little is known about their physiological impact. Here, the authors depart from the traditional standpoint and investigate their structure-function relationships. First, they show that spikes propagate more efficiently in axons with torpedoes. Secondly, they show that swellings can be induced pharmacologically and thirdly, a nice set of cerebellar learning experiments is used to correlate the torpedo number with learning. The findings are unexpected and, if true, represent an important conceptual advance. As detailed below, however, there are various concerns. Propagation fidelity is not really measured and a rational how swellings are mechanistically linked to learning is not presented.

1) The EM ultrastructure analysis in Figure 2 provides important new insights. There is an accumulation of mitochondria and neurofilaments are disrupted as expected from calcium-mediated degradation of the cytoskeleton. However, the authors make ambiguous remarks and details are lacking. "We identified 25 axons with diameters greater than 3 μm as putative axonal swellings". An axon is a swelling? Is the average axon diameter greater than 3 μm ? These numbers are not fitting with the Log2 x-axis in Figure 2d suggesting a range between 1 and 4 μm . What are the general properties of axons with and without torpedoes such as the internode diameter? This is a concern since in the legend it is written that "Axonal swellings are wrapped with moderately thicker myelin than neighboring control axons" but then in the Results they state; "we fortuitously found one instance ... which revealed was similar". At Page 6, lines 5 to 16 they claim that myelin around a swelling is thicker. This is impossible. If the entire internode diameter is larger it is expected that the oligodendroglial myelin sheath is thicker but myelin wrapping cannot be accumulated locally. An alternative and more likely explanation of the data in Figures 1 and 2 is that axons with torpedoes are larger in diameter. Larger diameters without a linear scaling with myelin thickness results in higher g-ratios (please note, axons with torpedoes have relatively less myelin as is shown in Fig. 2d). Differences in axonal diameters explain also the differences in failure. If the core axoplasmic resistance of an axon is reduced propagation is facilitated. The authors need to provide more concise description of the axon, the torpedo ultrastructure, including basic axon properties.

2) The data in Figure 1 represent important methodological advances and are of high quality. Unfortunately, like comment #1, the description of the Results is very limited and important details are lacking. What were the firing frequencies? What are the recordings distances from the swellings? Is there a cut-off frequency for failure rates? How robust are the failures from trial to trial, etc. etc.? How much different from random? The power of the analysis seems low and the authors are not controlling firing rates. They write (Line 9, p. 4) "Purkinje cell axons are high-fidelity conductors that display low propagation failure rates." It is rather the opposite; complex spikes evoked by climbing fibers are poorly conducted and filtered (Monsivais et al, J. Neurosci., 2005). For the arguments above, it is essential the authors experimentally control tonic firing rates and show that Purkinje cells with torpedoes are able to propagate higher frequencies. This is a simple experiment requiring somatic whole-cell and axonal loose-seal recordings. The climbing fiber plays an essential role in cerebellar learning by passing analog error signals and evoking complex spikes will teach us how these key signals are transferred and whether these are affected by torpedoes.

3) The experiments with bath application of nanomolar TTX (Figure 3) are creating lower probability of initiation but not a lower propagation. To provide evidence for a role of swellings in propagation

the authors will need to apply TTX locally to the axon.

4) What is the scientific rationale to use the cerebellar learning paradigms? The authors envision that firing rates reduce and more swellings appear? Do they postulate there is torpedo variation between subjects? There are remarks about structural plasticity occurring during learning in the discussion (line 11, p. 12) but without raising evidence how this can work. Subject-dependent differences are not evidence for structural plasticity. The learning experiments requires rewriting. A concern related to these data is that on multiple occasions the authors treat trends as being significant, e.g. Line 17, p 10 "Correlation ... mirrored the correlation ... although no significant difference". A P value of 0.052 is not different.

Minor

– Line 22, p. 3 "These data support a model in which Purkinje cell axonal swellings function to homeostatically preserve axonal function, rather than being pathophysiological." This is not supported by the data and is overselling the work.

– The fact that the authors observe that torpedoes in young mice (P10 to 3 months) are not detrimental to signal propagation does not imply they cannot play a pathological role later in life.

– The concept of homeostasis is not supported by the data. What is the optimal firing or failure rate?

– The authors should carefully scrutinize the manuscript for spelling. Just a very brief selection: p.29, line 1. "Close mitochondria", At p. 30, line 2 "Enhancinh"? At p. 31, line 10 "Indistinct". Line 7, p. 8 "As expect," > "As expected,"

– Why do the authors move away from the 'torpedo' nomenclature and refer to these as "axonal swelling"? They elegantly wrote in a previous related publication the term "torpedo" was used to identify these structures already in 1918. Since they are unique to Purkinje cells it seems wise to maintain consistency.

– Within Figure 2a there are two coloured boxed insets. Please clarify both.

– Figure 3d is analyzed with a two-way RM ANOVA?

– Page 35. "The density of axonal swellings in the flocculus is not significantly different in low and high VOR learners, perhaps due to a subset of swellings being labelled with IP3R, although the same trend is observed as for the Rotarod and Erasmus ladder tasks."

Differences are significant or they are not. If they are not and you are hesitating about them this means more data need to be collected.

Reviewer #2 (Remarks to the Author):

In this very interesting and surprising paper the authors argue that Purkinje neuron axons become

swollen to promote reliable and high fidelity axonal conduction. This is very surprising since these swellings have typically been thought of as pathological and indicative of injured or damaged axons. Thus, this paper significantly changes the way we think about these structures in the normal brain. In addition, it introduces a new form of axonal plasticity that has not been previously reported. As the authors point out, this doesn't exclude the possibility that these structures might also be pathological in other contexts. Nevertheless, the results presented are quite compelling and include excellent microscopic, pharmacological, and behavioral experiments that clearly describe the phenomenon. Overall, I think this is a very good paper that will be of great interest to neuroscientists.

I do have a few questions about the study and suggestions for a few improvements:

1. Page 3, line 23. "...rather than being pathophysiological." The authors make the point in the discussion that the swellings could still be pathophysiological in some settings. Therefore, they should give some qualifier like "...rather than only being pathophysiological." Because these swellings could be both!
2. I think the images (supplemental figure 1) with Caspr labeling could be improved, and I would even argue the data would be more compelling with nodal markers. In addition, from supplemental Fig. 1C, there appears to be a slight bias towards the swelling being after a node. But at least 40% occur before nodes. Wouldn't this tend to increase failures due to the large increase in capacitance? I understand how a node before the swelling might overcome the capacitive load, but not when the swelling precedes the node. Can the authors comment on why it is important to know the location of the swelling relative to a node. It would help to know how many instances were observed.
3. Page 12, line 15, "...neuroprotective..." I don't think the authors can or should conclude this phenomenon is neuroprotective. Instead, it may be compensatory, or adaptive to support neuronal function, but they have no evidence it is neuroprotective. In fact, in the context of some diseases it might promote neurodegeneration since this is a Ca²⁺-dependent phenomenon.
4. The authors have only investigated Purkinje neuron axons – can they comment on whether similar observations were made in other myelinated cell types? If not, what is special about Purkinje neurons that makes them need this kind of plasticity?
5. There are two big questions remaining: first, what molecular mechanisms related to Ca²⁺ induce the diameter change. I think this is an important question, but beyond the scope of this study. Second, the biggest question for me, and one that should at least be explored, is how does an increase in diameter promote more reliable transmission?

Reviewer #3 (Remarks to the Author):

The manuscript by Lang-Ouellette et al. reports unexpected findings that axonal swellings of cerebellar Purkinje cells (known as "torpedoes") are associated with enhanced fidelity of axonal conduction and cerebellar performance rather than pathological function. They performed dual extracellular/cell-attached recordings from the soma and the axon of the same Purkinje cells and found that conduction failure was reduced in axons with swellings. High levels of axonal conduction failures were mimicked by perfusing cerebellar slices with low concentration of TTX, which blocked action potentials in axons more preferentially than those in somata. This treatment led to formation of axonal swellings in cerebellar slices in 3 hours, which was blocked by removing extracellular Ca²⁺ or by adding Ni²⁺, suggesting the involvement of T-type voltage-dependent Ca²⁺ channels.

Furthermore, the authors performed three types of cerebellum-dependent motor learning tasks and showed that mice with higher level of learning had higher number of axonal swellings. They propose that axonal swellings in Purkinje cells underlie a form of neural plasticity that optimizes the fidelity of action potential propagation and enhances cerebellar motor learning.

In general, the experiments were well performed, the data are clear and nicely presented. The results are very interesting and may attract readers of various neuroscience fields. However, I have several concerns as listed below.

Major points

1) The authors claimed that “low TTX (10 nM)” caused higher degree of action potential blockade in the axon than in the soma (Figure 3a and 3b), thus mimicking high axonal conduction failures. However, they do not show the number of Purkinje cells examined in this experiment. What do the error bars represent in Figure 3b? There is a slight difference between the firing frequency at the soma and that at the axon from 6 min to 15 min but there seemed to be no difference between the two after 30 min (Figure 3b). Since the authors showed that treatment with “low TTX (10 nM)” for 30 min followed by “high TTX (200 nM)” for 2.5 hours did not induce new axonal swellings, it is important to show more quantitatively to what extent and how long axonal conduction failure persists after the initiation of “low TTX (10 nM)” perfusion.

2) I think it also necessary to show the time course of action potential blockade at the soma and the axon by “high TTX (200 nM)”. Are action potentials at the soma and the axon blocked completely?

3) The authors suggest that Ca²⁺ entry through T-type voltage-dependent Ca²⁺ channel is required for the “low TTX”-induced formation of new axonal swellings. This argument is based on the results that the formation of new axonal swellings was blocked by removal of extracellular Ca²⁺ or by adding high concentration of Ni²⁺. Although 1 mM Ni²⁺ was considered to block all types of T-type Ca²⁺ channels, I am concerned that Ni²⁺ might have unexpected side effects. Therefore I feel evidence is not strong enough to conclude the involvement of T-type Ca²⁺ channel. I suggest the authors test whether other types of voltage-dependent Ca²⁺ channels (P/Q-type, L-type...) are involved by using specific blockers.

4) If Ca²⁺ entry through T-type Ca²⁺ channels triggers the formation of axonal swelling, it is expected that Ca²⁺ transients can be detected at the location of new axonal swelling. The authors can perform Ca²⁺ imaging from axons of Purkinje cells that express a GECI such as GCaMP6 during “low TTX”-induced formation of new axonal swellings.

5) The positive correlation between the density of axonal swellings and “the amount of motor learning (i.e., [% Motor performance at Day 1] – [% Motor performance at Day 7])” is very interesting. How about the relationship between the density of axonal swellings and “% Motor performance at Day 8”? Are these two parameters positively correlated? If not, the data suggest that the density of axonal swellings does not correlate with the state of motor performance but reflects the dynamic change in the level of motor performance during learning.

Minor points

1) Figure 4: In experiments with “0 Ca²⁺”, “3 mM Ca²⁺” or “1 mM Ni²⁺ + low TTX”, , there is no description about Mg²⁺ concentration in the external solutions. Did the authors keep the total divalent ion concentration constant and change Mg²⁺ concentration accordingly?

2) In the context of plasticity of axon initial segment, the papers by Kuba et al. (Nature 444 (7122), 1069-1072; Nature communications 6 (1), 1-12) should be cited.

Reviewer #4 (Remarks to the Author):

The study addresses the role of morphological swellings of axons of Purkinje cells for cellular and behavioral functions of the cerebellum. They show that the presence of the swellings is correlated with reduced failures of AP propagation recorded along Purkinje cell axons in acute brain slices and with improved performance on several cerebellar-related behavioural assays. They also show that pharmacological conditions that enhance AP propagation failures (but not AP generation per se) induce the formation of new swellings. They conclude that these swellings are beneficial for cerebellar function by reducing failures of AP propagation. As this effect can only be rather modest because the failure rate in axons without swellings is already pretty low, the case seems not entirely convincing.

But still, the study presents some interesting new data, and is generally well executed and written. I have these comments / suggestions for the authors to consider to try to improve their study and make it more nutritious.

- The swellings are correlated with reduced AP propagation failures, but the underlying mechanisms remain unclear. The biophysics of voltage spread would predict the opposite because of the effect of extra membrane capacitance from the swellings.
- Are there any measurable differences in conduction delay? The swellings should slow down conduction. They could normalize delay to axon length.
- The TTX effects offer the chance to do a before-after experiment and correlate the formation of a new swelling with AP propagation. This would provide very strong evidence.
- What is the fraction of axons that have zero, one or multiple swellings? Or did I miss this essential info? Is propagation correlated with the number of swellings (if more than one exist)?
- Purkinje cells occasionally fire complex spikes. Is their propagation also correlated with the swellings?

Response to Reviewers for Lang-Ouellette et al., NCOMMS-20-05788

Reviewer #1 (Remarks to the Author):

The manuscript by Lang-Ouellette and colleagues presents a very exciting set of results. The so-called torpedoes in Purkinje cell axons have been topic of investigation in the context of pathology for several decades but little is known about their physiological impact. Here, the authors depart from the traditional standpoint and investigate their structure-function relationships. First, they show that spikes propagate more efficiently in axons with torpedoes. Secondly, they show that swellings can be induced pharmacologically and thirdly, a nice set of cerebellar learning experiments is used to correlate the torpedo number with learning. The findings are unexpected and, if true, represent an important conceptual advance. As detailed below, however, there are various concerns. Propagation fidelity is not really measured and a rational how swellings are mechanistically linked to learning is not presented.

We thank the reviewer for their positive comments about our results and manuscript and for their careful reading of our manuscript, and useful and insightful feedback. We have addressed the concerns raised by Reviewer 1 below and in our revised manuscript.

1) The EM ultrastructure analysis in Figure 2 provides important new insights. There is an accumulation of mitochondria and neurofilaments are disrupted as expected from calcium-mediated degradation of the cytoskeleton. However, the authors make ambiguous remarks and details are lacking. “We identified 25 axons with diameters greater than 3 μm as putative axonal swellings”. An axon is a swelling? Is the average axon diameter greater than 3 μm ? These numbers are not fitting with the Log2 x-axis in Figure 2d suggesting a range between 1 and 4 μm .

We thank the reviewer for their careful read of our paper and thoughtfulness about our data. While we had based our non-swelling axonal diameter on fluorescent microscopy data where we seldom observed axons with diameters $> 3 \mu\text{m}$ (Ljungberg *et al.*, 2016), our EM data where we have axon swellings and axons connected suggests that in some cases, axon diameter can be as large as up to 4 μm . This means that the EM images we have collected with axon diameters in the 2 – 4 μm range likely include a mixed population of smaller axonal swellings and thicker axons, which we worried might be skewing our findings. To ensure that our analysis of our EM data is as representative as possible of distinct categories, we have reanalyzed our data using a stricter $> 4 \mu\text{m}$ threshold for axonal swellings, and have excluded all axons in the 2 – 4 μm range from our control axon and axonal swelling categories (since both are likely included in this set of data), and have a maximum of up to 2 μm for axons. In this way, we should be comparing axons to axonal swellings without ambiguity. The general findings that we have reported for our data (for e.g. g-ratio) were not significantly different using this criteria than with our previous analysis. We have additionally added new analysis of organelle density, which we find is not significantly different between axons and swellings.

What are the general properties of axons with and without torpedoes such as the internode diameter?

We thank the reviewer for highlighting this omission from the paper. We have measured the axonal diameters from axons with axonal swellings and neighboring axons without using 2-photon microscopy of GFP-filled axons analyzed with NeuroLucida. We find that there are no significant differences in axonal diameter when comparing axons with swellings (excluding the swellings) and axons without swellings. Since nodes are only a small proportion of the total axonal length, these measurements largely reflect internodal diameter. This data is now included in Supplementary Fig. 6. We have also included additional properties of axons in Supplementary Table 1.

This is a concern since in the legend it is written that “Axonal swellings are wrapped with moderately thicker myelin than neighboring control axons” but then in the Results they state; “we fortuitously found one instance ... which revealed was similar”. At Page 6, lines 5 to 16 they claim that myelin around a swelling is thicker. This is impossible. If the entire internode diameter is larger it is expected that the oligodendroglial myelin sheath is thicker but myelin wrapping cannot be accumulated locally. An alternative and more likely explanation of the data in Figures 1 and 2 is that axons with torpedoes are larger in diameter. Larger diameters without a linear scaling with myelin thickness results in higher g-ratios (please note, axons with torpedoes have relatively less myelin as is shown in Fig. 2d).

We apologize for any confusion that this statement made. We did not mean to imply that the myelin around axonal swellings was thicker than that around flanking axons. We are unable to address this, since we do not have serial reconstructions of swellings and flanking axons, and measuring both the axon and axonal swelling on the same axon is a rare event in EM, which we only observed twice. In those cases, the myelin thickness was similar around the axon and the swelling. On average, however, we find a modest increase in myelin thickness if we simply compare the myelin around axonal swellings and control axons ($< 2 \mu\text{m}$). We have made our measurements more stringent, and only measure the images where the myelin is best preserved in the EM images, which explains our lower n for these measurements. While we have measured this the best we can, we acknowledge that these findings are difficult to make with a limited EM dataset, and thus have moved these findings to a Supplementary figure (Supplementary Fig. 6). However, the major finding that myelin thickness does not scale with diameter of the axonal swellings, resulting in significantly higher g-ratios for axonal swellings than for control axons is robust, and we have kept this in the main figure (Fig. 2d).

Differences in axonal diameters explain also the differences in failure. If the core axoplasmic resistance of an axon is reduced propagation is facilitated. The authors need to provide more concise description of the axon, the torpedo ultrastructure, including basic axon properties.

We thank the reviewer for this important suggestion, which we agree we should address in more depth. To do this, we have now measured the density of organelles in both non-swelling axons and in axonal swellings. We report this data in Fig. 2c. We have also measured the diameter of axons with and without swellings, which we report in Supplementary Fig. 6, and summarize these findings in a table (Supplementary Table 1). We also include a new section in the Discussion (from page 13 line 17, to page 14 line 15) where we discuss how and if the biophysical properties of the axonal swellings may influence propagation.

An alternative and more likely explanation of the data in Figures 1 and 2 is that axons with torpedoes are larger in diameter.

This is an interesting suggestion, and we thank the reviewer for their input. We also wondered about this possibility. To test this, we measured axonal diameter from axons with and without swellings from 2-photon image stacks of GFP-expressing axons using Neurolucida, and found that there was no significant difference between their diameters, suggesting that this was not likely (Supplementary Fig. 6a). However, measuring axon diameter from EM images would give us the most accurate measurement of diameter, and it would be ideal to compare axons and swellings from the same structure. Unfortunately, given the limitations of EM, and the rareness of swellings, we are unable to test this directly in our EM images, since we only observe swellings with connected flanking axons only in 2 images. These axons are $> 2 \mu\text{m}$, which is seen in Supplementary Fig. 6 (that is, on the large side for axons), and the myelin thickness was very similar around the swelling and the flanking axon, supporting the Reviewer's suggestion. However, when we systematically analyze the axonal diameters from flanking axons in fluorescent images (Supplementary Fig. 6a), we find no significant differences.

2) The data in Figure 1 represent important methodological advances and are of high quality. Unfortunately, like comment #1, the description of the Results is very limited and important details are lacking. What were the firing frequencies?

We thank the Reviewer for their positive and helpful comments. Firing frequencies are not statistically different, and are included in Supplementary Fig. 3 and are also reported in Supplementary Table 1.

What are the recordings distances from the swellings?

Recording distances are reported in Supplementary Table 1 and are shown in Supplementary Fig. 2c. Recording distances for axons and swellings are not significantly different.

Is there a cut-off frequency for failure rates?

We observed no relationship between frequency and failure rate, shown in Supplementary Fig. 3, for either control axons or axons with swellings. This suggests that although axons fail more when driven to spike at high frequencies (e.g. Khaliq et al., 2005; Monsivais et al., 2005; Hirono et al., 2015), there appears to be no similar relationship between baseline firing rate and axonal failure, suggesting that baseline failure rate is set by a distinct mechanism to that which causes failures at high frequencies. We discuss this directly in the Results section (page 5, lines 5-14).

How robust are the failures from trial to trial, etc. etc.? How much different from random?

We show failure rate as a function of time for each cell in Supplementary Fig. 2. Cells that had high propagation failure rates tended to have high failure rates throughout the recording, and cells

that had low propagation failures tended to have low failure rates. The duration of the recordings differed from paired recording to paired recording, but were not different across condition.

The power of the analysis seems low and the authors are not controlling firing rates. They write (Line 9, p. 4) “Purkinje cell axons are high-fidelity conductors that display low propagation failure rates.” It is rather the opposite; complex spikes evoked by climbing fibers are poorly conducted and filtered (Monsivais et al, J. Neurosci., 2005).

We apologize for lack of clarity. We have rewritten this section to better reflect the literature, and have included a new section in the Discussion focused on the important question of complex spike propagation on page 15, line 21 to page 16, line 2.

For the arguments above, it is essential the authors experimentally control tonic firing rates and show that Purkinje cells with torpedoes are able to propagate higher frequencies. This is a simple experiment requiring somatic whole-cell and axonal loose-seal recordings. The climbing fiber plays an essential role in cerebellar learning by passing analog error signals and evoking complex spikes will teach us how these key signals are transferred and whether these are affected by torpedoes.

We appreciate the reviewer’s interesting interpretation of this data. However, since we find no relationship between firing rate and failure (Supplementary Fig. 3), we are careful to not claim that failure rate and firing frequency are linked (page 5, lines 5-14). While it would be interesting to see if evoked high-frequency axonal failures and evoked climbing fiber transmission was altered in the presence of axonal swellings, this is a separate question to what we have posed in this manuscript. We feel that this would be better suited to a follow-up study rather than this initial characterization of axonal failure rates with swellings.

3) The experiments with bath application of nanomolar TTX (Figure 3) are creating lower probability of initiation but not a lower propagation. To provide evidence for a role of swellings in propagation the authors will need to apply TTX locally to the axon.

We observe significantly fewer axon action potentials than somatic action potentials in *low TTX*, which suggests to us that propagation is affected. There is also lower initiation, which explains why the firing rates drop in *low TTX*. We favor the explanation that it is the difference in action potentials between the soma and the axon that give rise to the formation of axonal swellings, thus a chemical “mimicking” of axonal failures. Another way to show this would be by disrupting axonal propagation focally, such as with local TTX application. However, our data using *low TTX* allows us to treat many axons simultaneously. Since we observe that only a subset of axons form swellings, this would likely be an extremely low-yield experiment. We have not been able to determine which axons will form swellings and which do not, and our data suggests that there is no morphological signature, since their diameters are not different at time 0 (Fig. 3g-j). Thus, bath application of *low TTX* allows us to manipulate activity in multiple Purkinje cells and increase the yield of these experiments. To look in more depth into the amount of axonal failures needed to create a swelling, we have included a new paradigm where we bath apply low TTX for 1.5 hours after 0.5 hours of high TTX, and find that this is not sufficient to cause axonal swellings to form.

This data is now included in Supplementary Fig. 7, and in the Results section on page 8, line 23 to page 9, line 5. These data argue that a minimum number of axonal failures is required to form swellings.

4) What is the scientific rationale to use the cerebellar learning paradigms?

We thank the reviewer for highlighting this confusion, and have rewritten the Discussion sections to better explain the rationale (page 15, lines 5-11). We wanted to use a measurement that would allow us to compare across different cerebellar assays. We state more clearly that we do not think that swellings have a direct effect on learning; rather, it likely occurs due to improved overall cerebellar performance arising from enhanced action potential propagation fidelity in the axon, and is thus likely indirect.

The authors envision that firing rates reduce and more swellings appear? Do they postulate there is torpedo variation between subjects?

We thank the reviewer for their comment, and apologize for lack of clarity. We do in fact observe a great deal of animal-to-animal variation across mice, which is shown in Figure 5b, g, and l. Indeed, our early preliminary observation that variability in the axonal swelling density existed across animals was one of the driving forces for these experiments. We have also explained our rationale (as described above, page 15, lines 9-17).

There are remarks about structural plasticity occurring during learning in the discussion (line 11, p. 12) but without raising evidence how this can work. Subject-dependent differences are not evidence for structural plasticity. The learning experiments requires rewriting. A concern related to these data is that on multiple occasions the authors treat trends as being significant, e.g. Line 17, p 10 "Correlation ... mirrored the correlation ... although no significant difference". A P value of 0.052 is not different.

We thank the reviewer for these comments. We have corrected this section to clearly state that a positive but non-significant correlation is observed (page 11, line 18 to page 12, line 9), and that this suggests that axonal swellings in the flocculus cannot account for much of the variability in learning across animals for VOR adaptation.

Minor

– Line 22, p. 3 "These data support a model in which Purkinje cell axonal swellings function to homeostatically preserve axonal function, rather than being pathophysiological." This is not supported by the data and is overselling the work.

We thank the reviewer for their input and have rewritten this section in response to this comment (page 3, line 22-23).

– The fact that the authors observe that torpedoes in young mice (P10 to 3 months) are not detrimental to signal propagation does not imply they cannot play a pathological role later in life.

We agree with this observation and have rewritten Discussion to more clearly reflect this (page 16, line 12 to page 17 line 3).

– The concept of homeostasis is not supported by the data. What is the optimal firing or failure rate?

We thank the reviewer for their comment. We have rewritten this section to better reflect this and to improve clarity (page 14, lines 16-22). We are not arguing for an optimal firing rate for Purkinje cells. However, we feel that a brief discussion of homeostasis is still meaningful, because torpedo formation appears to optimize axonal function, by reducing axonal failures, thereby achieving an optimal propagation level. This is distinct from other forms of homeostasis such as firing rate homeostasis or synaptic scaling, but nonetheless still appears to serve an adaptive function.

– The authors should carefully scrutinize the manuscript for spelling. Just a very brief selection: p.29, line 1. “Close mitochondria”, At p. 30, line 2 “Enhancinh”? At p. 31, line 10 “Indistinct”. Line 7, p. 8 “As expect,” > “As expected,”

We thank the reviewer for these comments, and have checked the manuscript carefully for spelling and formatting errors.

– Why do the authors move away from the ‘torpedo’ nomenclature and refer to these as “axonal swelling”? They elegantly wrote in a previous related publication the term “torpedo” was used to identify these structures already in 1918. Since they are unique to Purkinje cells it seems wise to maintain consistency.

We thank the reviewer for these comments. We chose to reword this because there are differences in the field about what constitutes a torpedo and what does not, and they have typically been used to describe disease-related swellings rather than those observed in healthy brains. We thus use the general term “axonal swelling” so that we are as clear as possible and do not confuse readers who have a different understanding of an axonal torpedo.

– Within Figure 2a there are two coloured boxed insets. Please clarify both.

We thank the reviewer for noticing this confusion, which we apologize was a formatting error that crept into the image. We have removed this.

– Figure 3d is analyzed with a two-way RM ANOVA?

We have reanalyzed all our time-lapse imaging using repeat-measure ANOVAs, and we are now reporting the interaction effect of all time-lapse imaging using a Greenhouse-Geisser correction, to correct for non-sphericity of the data. Post-hoc analysis with Bonferroni correction was subsequently done. Furthermore, to better represent the different conditions, we have added bar graphs for each experiment allowing a comparison of axonal swellings after 3 hours. For this

analysis, a Kruskal-Wallis H test (non-parametric one-way ANOVA) was used, followed by Mann-Whitney U tests with Bonferroni correction. These values have been updated in figures, figure legends, and in the text.

– Page 35. “The density of axonal swellings in the flocculus is not significantly different in low and high VOR learners, perhaps due to a subset of swellings being labelled with IP3R, although the same trend is observed as for the Rotarod and Erasmus ladder tasks.”

Differences are significant or they are not. If they are not and you are hesitating about them this means more data need to be collected.

We thank the reviewer for his/her comments. We have rewritten our Results section to change this (page 11, line 18 to page 12, line 9).

Reviewer #2 (Remarks to the Author):

In this very interesting and surprising paper the authors argue that Purkinje neuron axons become swollen to promote reliable and high fidelity axonal conduction. This is very surprising since these swellings have typically been thought of as pathological and indicative of injured or damaged axons. Thus, this paper significantly changes the way we think about these structures in the normal brain. In addition, it introduces a new form of axonal plasticity that has not been previously reported. As the authors point out, this doesn't exclude the possibility that these structures might also be pathological in other contexts. Nevertheless, the results presented are quite compelling and include excellent microscopic, pharmacological, and behavioral experiments that clearly describe the phenomenon. Overall, I think this is a very good paper that will be of great interest to neuroscientists.

We thank the reviewer for their positive comments about our results and manuscript and for their careful reading of our manuscript, and useful and insightful feedback. We have addressed the concerns raised by Reviewer 2 below and in our revised manuscript.

I do have a few questions about the study and suggestions for a few improvements:

1. Page 3, line 23. "...rather than being pathophysiological." The authors make the point in the discussion that the swellings could still be pathophysiological in some settings. Therefore, they should give some qualifier like "...rather than only being pathophysiological." Because these swellings could be both!

We thank the reviewer for this comment and suggestion. We have re-written this section for clarity, and have removed "rather than being pathophysiological" entirely on page 3. However, we also address this in more depth in the Discussion (page 16, line 12 to page 17, line 3).

2. I think the images (supplemental figure 1) with Caspr labeling could be improved, and I would even argue the data would be more compelling with nodal markers.

We thank the reviewer for this insightful comment. We agree that nodal markers would have been ideal. We have tried a number of different nodal labels including Ankyrin G, Nav1.6, and β -spectrinIV, and found that the Caspr labeling was the most robust by far. Unfortunately, we were unable to get good quality data from other antibodies. Several other research groups have utilized Caspr labeling to mark perinodal structures in Purkinje cell axons (e.g. Hirono et al., J. Neurosci., 2015; Barron et al., Scientific Reports, 2018), supporting this marker as a label for perinodal structures. We have rewritten the Results to make clear the limitations presented by perinodal labeling (page 6, lines 3-5). We have also worked to improve our images, by changing the look-up tables. These new Caspr images are shown in Supplementary Fig. 4.

In addition, from supplemental Fig. 1C, there appears to be a slight bias towards the swelling being after a node. But at least 40% occur before nodes. Wouldn't this tend to increase

failures due to the large increase in capacitance? I understand how a node before the swelling might overcome the capacitive load, but not when the swelling precedes the node. Can the authors comment on why it is important to know the location of the swelling relative to a node. It would help to know how many instances were observed.

We thank the reviewer for these interesting comments, and have indicated how many nodes were analyzed now in the figure legend of Supplementary Fig. 4 (n = 58 swellings). We have measured the location of the node relative to the swelling from the Purkinje cell layer, but have not traced the axons in every case, meaning that it is possible that the swelling occurs on a recurrent collateral and thus the orientation is reversed (although this would expect to be occur relatively infrequently). How axonal swelling location relative to nodes affects propagation from a biophysical point of view is challenging to interpret. We have included an expanded section on this in the Discussion (page 13, line 17 to page 14, line 15). We agree that it warrants further investigation, likely through a detailed biophysical model, but that it is beyond the scope of this paper.

3. Page 12, line 15, "...neuroprotective..." I don't think the authors can or should conclude this phenomenon is neuroprotective. Instead, it may be compensatory, or adaptive to support neuronal function, but they have no evidence it is neuroprotective. In fact, in the context of some diseases it might promote neurodegeneration since this is a Ca²⁺-dependent phenomenon.

We thank the reviewer for this comment. We wrote "neuroprotective" as a synonym for adaptive or compensatory, but we realize that this definition of neuroprotective may be erroneous. We have rewritten this section to make our meaning clearer.

4. The authors have only investigated Purkinje neuron axons – can they comment on whether similar observations were made in other myelinated cell types? If not, what is special about Purkinje neurons that makes them need this kind of plasticity?

This is an interesting question. There are several other myelinated axons that display axonal swellings. While there may be reasons that Purkinje cells have a special requirement for axonal swellings, it is also possible that similar swellings are observed on other cell types. We have added a section addressing this in the Discussion (page 16, lines 3-11).

5. There are two big questions remaining: first, what molecular mechanisms related to Ca²⁺ induce the diameter change. I think this is an important question, but beyond the scope of this study. Second, the biggest question for me, and one that should at least be explored, is how does an increase in diameter promote more reliable transmission?

We agree with the reviewer that the question of the molecular mechanism is an important question, but that it is unfortunately beyond the scope of the paper. To address how an increase in diameter could promote more reliable transmission, we have added a section to the Discussion addressing this (page 13, line 17 to page 14, line 15), and have added additional analysis of organelle density to look more in depth at the biophysical properties of axonal swellings and to relate them to axonal propagation (Fig. 2c and Supplemental Table 1).

Reviewer #3 (Remarks to the Author):

The manuscript by Lang-Ouellette et al. reports unexpected findings that axonal swellings of cerebellar Purkinje cells (known as “torpedoes”) are associated with enhanced fidelity of axonal conduction and cerebellar performance rather than pathological function. They performed dual extracellular/cell-attached recordings from the soma and the axon of the same Purkinje cells and found that conduction failure was reduced in axons with swellings. High levels of axonal conduction failures were mimicked by perfusing cerebellar slices with low concentration of TTX, which blocked action potentials in axons more preferentially than those in somata. This treatment led to formation of axonal swellings in cerebellar slices in 3 hours, which was blocked by removing extracellular Ca²⁺ or by adding Ni²⁺, suggesting the involvement of T-type voltage-dependent Ca²⁺ channels. Furthermore, the authors performed three types of cerebellum-dependent motor learning tasks and showed that mice with higher level of learning had higher number of axonal swellings. They propose that axonal swellings in Purkinje cells underlie a form of neural plasticity that optimizes the fidelity of action potential propagation and enhances cerebellar motor learning.

In general, the experiments were well performed, the data are clear and nicely presented. The results are very interesting and may attract readers of various neuroscience fields. However, I have several concerns as listed below.

We thank the reviewer for their positive comments about our results and manuscript and for their careful reading of our manuscript, and useful and insightful feedback. We have addressed the concerns raised by Reviewer 3 below and in our revised manuscript.

Major points

1) The authors claimed that “low TTX (10 nM)” caused higher degree of action potential blockade in the axon than in the soma (Figure 3a and 3b), thus mimicking high axonal conduction failures. However, they do not show the number of Purkinje cells examined in this experiment.

We thank the reviewer for this comment and apologize for this omission, which we have corrected in both the text and figure legends (n = 17 paired recordings).

What do the error bars represent in Figure 3b? There is a slight difference between the firing frequency at the soma and that at the axon from 6 min to 15 min but there seemed to be no difference between the two after 30 min (Figure 3b). Since the authors showed that treatment with “low TTX (10 nM)” for 30 min followed by “high TTX (200 nM)” for 2.5 hours did not induce new axonal swellings, it is important to show more quantitatively to what extent and how long axonal conduction failure persists after the initiation of “low TTX (10 nM)” perfusion.

We thank the reviewer for this comment. The error bars reflect changes across cells, as we recorded from 17 pairs of soma and axons. These recordings were in many cases not arising from the same cell (that is, they were simultaneous soma/axon recordings from neighboring cells), but which

nonetheless allowed us to observe the relative time course of spiking activity in different compartments of different cells simultaneously. We agree that understanding the extent and duration of persistent axonal failures that causes swelling formation is important. To understand this better, we include new data in Supplementary Fig. 7, where *high TTX* was perfused for 30 minutes followed by *low TTX* for 1.5 hours. Surprisingly, no new swellings formed, despite *low TTX* being present 1.5 hours, but without the relatively slow onset of failures that would have been observed when perfusing *low TTX* directly. This suggests that a critical amount of axonal failure is required for the formation of axonal swellings, that is not achieved with either of these protocols.

2) I think it also necessary to show the time course of action potential blockade at the soma and the axon by “high TTX (200 nM)”. Are action potentials at the soma and the axon blocked completely?

We thank the reviewer for this comment. Our *high TTX* concentration (200 nM) is double what have been used before to block Na⁺ current in Purkinje cells. Others have estimated that a concentration as little as 100nM would be approximately ~30X the K_d for binding to Na⁺ channels, and should thus be fully saturating at these ages (Khaliq and Raman, J. Neurosci., 2006). While the time course of this blockade has not been determined by us, others have reported that it can block very rapidly, within 100 ms when applied directly to the cell. Since we are bath perfusing TTX, our time course will be largely determined by the flowrate of our perfusion. We expect *high TTX* will be fully perfused within ~5-10 minutes, and that action potentials will be fully blocked at that time. Indeed, this agrees with our finding that we see a change in firing frequency in *low TTX* after 5-6 minutes. We expect *low TTX* to be slower as accumulation of blocked Na⁺ channels removing the pool of Na⁺ channels available to participate in the action potential in this case is likely to take time. While we would like to confirm the time course of blockade in *high TTX* with our own hands, we are unable to acquire these experiments due to our laboratory being shut down by the COVID-19 pandemic. However, since we see a difference between *high* and *low* TTX, we can surmise that there is indeed a difference, and we feel that although they would be interesting, these results would not significantly alter or improve upon our findings.

3) The authors suggest that Ca²⁺ entry through T-type voltage-dependent Ca²⁺ channel is required for the “low TTX”-induced formation of new axonal swellings. This argument is based on the results that the formation of new axonal swellings was blocked by removal of extracellular Ca²⁺ or by adding high concentration of Ni²⁺. Although 1 mM Ni²⁺ was considered to block all types of T-type Ca²⁺ channels, I am concerned that Ni²⁺ might have unexpected side effects. Therefore I feel evidence is not strong enough to conclude the involvement of T-type Ca²⁺ channel. I suggest the authors test whether other types of voltage-dependent Ca²⁺ channels (P/Q-type, L-type...) are involved by using specific blockers.

We agree that the current evidence does not rule out other channels. Thus, we have re-written this section to better indicate that other voltage-dependent calcium channels might also play a role (page 10, lines 8-9; page 10, line 15). We also include data showing the effect of a sub-saturation concentration of nickel (100 μM) which is equivalent to the IC₅₀ of most T-type calcium channels in Supplementary Fig. 8 and in the Results (page 10, lines 12-15). This shows an intermediate reduction (~35%) of the formation of axonal swellings.

4) If Ca²⁺ entry through T-type Ca²⁺ channels triggers the formation of axonal swelling, it is expected that Ca²⁺ transients can be detected at the location of new axonal swelling. The authors can perform Ca²⁺ imaging from axons of Purkinje cells that express a GECI such as GCaMP6 during “low TTX”-induced formation of new axonal swellings.

We thank the reviewer for this interesting comment. We agree that the current evidence suggests that calcium transients would be observed in axonal segments. While we would like to explore calcium dynamics in swellings directly, we are unable to complete these experiments at this time, and feel that these experiments are not crucial to the major findings of the paper. Given the slow time course of swelling formation, it is possible that detection happen elsewhere (for example, in the axon initial segment) and then are transported down the axon to form a swelling. This has been addressed in more detail in the Discussion section (page 15 lines 3-8).

5) The positive correlation between the density of axonal swellings and “the amount of motor learning (i.e., [% Motor performance at Day 1] – [% Motor performance at Day 7])” is very interesting. How about the relationship between the density of axonal swellings and “% Motor performance at Day 8”? Are these two parameters positively correlated? If not, the data suggest that the density of axonal swellings does not correlate with the state of motor performance but reflects the dynamic change in the level of motor performance during learning.

This is an interesting idea, which we have examined by re-analyzing our data. We found that there was a similar positive correlation between motor performance on the last day and the number of axonal swellings, although typically not as strong a correlation as observed with learning. It is difficult to disentangle motor performance from motor learning in our hands, as enhanced motor performance is likely to lead to enhanced learning. We have addressed this more directly in the Discussion section (page 15 lines 9-20).

Minor points

1) Figure 4: In experiments with “0 Ca²⁺”, “3 mM Ca²⁺” or “1 mM Ni²⁺ + low TTX”, , there is no description about Mg²⁺ concentration in the external solutions. Did the authors keep the total divalent ion concentration constant and change Mg²⁺ concentration accordingly?

We thank the reviewer for this comment. We did not change the Mg²⁺ concentrations and have indicated this in the Material and Methods (page 20, lines 21-23).

2) In the context of plasticity of axon initial segment, the papers by Kuba et al. (Nature 444 (7122), 1069-1072; Nature communications 6 (1), 1-12) should be cited.

We thank the reviewer for noticing these omissions, and have added these citations to the manuscript (citations 50 and 51).

Reviewer #4 (Remarks to the Author):

The study addresses the role of morphological swellings of axons of Purkinje cells for cellular and behavioral functions of the cerebellum. They show that the presence of the swellings is correlated with reduced failures of AP propagation recorded along Purkinje cell axons in acute brain slices and with improved performance on several cerebellar-related behavioural assays. They also show that pharmacological conditions that enhance AP propagation failures (but not AP generation per se) induce the formation of new swellings. They conclude that these swellings are beneficial for cerebellar function by reducing failures of AP propagation. As this effect can only be rather modest because the failure rate in axons without swellings is already pretty low, the case seems not entirely convincing.

But still, the study presents some interesting new data, and is generally well executed and written. I have these comments / suggestions for the authors to consider to try to improve their study and make it more nutritious.

We thank the reviewer for their positive comments about our results and manuscript and for their careful reading of our manuscript, and useful and insightful feedback. We have addressed the concerns raised by Reviewer 4 below and in our revised manuscript.

- The swellings are correlated with reduced AP propagation failures, but the underlying mechanisms remain unclear. The biophysics of voltage spread would predict the opposite because of the effect of extra membrane capacitance from the swellings.

We thank the reviewer for this comment. We agree that biophysical modeling of voltage spread would not necessarily agree with our findings, and have discussed how the biophysics of axonal swellings might influence propagation in more depth in the Discussion section (page 13, line 17 to page 14 line 15). Our interpretation of our results is that enhanced propagation likely arises from intracellular signaling cascades in the swellings rather than biophysical properties of the swellings.

- Are there any measurable differences in conduction delay? The swellings should slow down conduction. They could normalize delay to axon length.

We thank the reviewer for noting this omission, and have reported the conduction delay and speed of propagation in Supplementary Fig. 2b. We observe no significant differences in conduction velocity, although we note that we are measuring over relatively short distances, which may mean that we cannot detect a small change. We report this in the Results section (page 4, line 21 to page 5, line 4) and discuss these findings further in the Discussion (page 13, line 17 to page 14, line 4).

- The TTX effects offer the chance to do a before-after experiment and correlate the formation of a new swelling with AP propagation. This would provide very strong evidence.

We thank the reviewer for this comment, which we agree would be a very interesting experiment. We have been unable to successfully perform this experiment, however, because of technical challenges: it is extremely difficult to maintain a dual somatic/axonal recording over the time

course (> 2 hours) that we would need to in order to observe a swelling form. While such a finding would provide further evidence, we feel that our findings, using different paradigms (e.g. inclusion of new data in Supplementary Fig. 7), and different pharmacological approaches (e.g. Figs. 3 and 4), provide strong evidence supporting our findings that axonal failures lead to the formation of axonal swellings.

What is the fraction of axons that have zero, one or multiple swellings? Or did I miss this essential info? Is propagation correlated with the number of swellings (if more than one exist)?

We thank the reviewer for noticing this omission. We have previously published that ~30% of axons at this age have swellings (Ljungberg et al., 2016). The vast majority (98.7%, or 950/963) of axons with swellings had single swellings in the granule cell layer, which are the ones that we targeted. We have included this important information in Supplementary Fig. 1, and cite our earlier paper in the Results (page 4, lines 11-14) and Methods section (page 20, lines 8-10) showing the overall proportion. Since we have recording exclusively in axons with single swellings, we cannot comment on whether propagation correlates with the number of swellings. Although it is an intriguing question, the infrequency of multiple swellings on axons means it would be challenging to determine.

- Purkinje cells occasionally fire complex spikes. Is their propagation also correlated with the swellings?

We thank the reviewer for this comment, and agree that understanding how complex spike propagation is influenced by axonal swellings is an interesting question that we have not addressed. However, we have added a section in the Discussion where we discuss this important question (page 15, line 21 to page 16, line 2).

Reviewer #1 (Remarks to the Author):

In this revision the authors clarified some of the previous concerns by reanalyzing and re-plotting data together with textual changes. New data is shown in the supplement (although this would be better placed in the main results). While the topic still has broad interest and some aspects became clearer, this revision dampened my enthusiasm. The key question how a swelling alters Purkinje axon transmission fidelity remains unanswered and new errors have been introduced.

#1A Regarding the EM, the claim (p. 14, line 261-264) of an increased myelin thickness underlying an enhanced action potential fidelity is not supported by the data. A g-ratio is not necessarily reflecting the myelin thickness. The g-ratio is the fraction of outer fiber diameter/axon diameter. Both can be different and not necessarily proportional. Showing a higher g-ratio thus leaves open various possibilities and Supplemental Figure 6 is required to interpret Figure 2. The authors need to show g-ratio as a function of linear axon diameter (removing also their error of the log₂ label). This reviewer plotted this graph based on the source data and the relationship suggests a continuum with larger axons having an increasingly thinner sheath. Such relationship is expected for the large diameter range between 1 and 10 μm (see e.g. Berthold and Rydmark, 1983). The only conclusion which can be drawn is that the swelling is not a synaptic terminal and the myelin sheath is relatively thinner but the largest difference is the axon core diameter (which is logic since the authors also selected for this parameter). There is a discrepancy with the functional data since if the myelin sheath is thicker across the entire internode the velocity should be higher (if measured over sufficiently large distances).

#1B. There is no increased velocity (Suppl Fig. 2d). However, something seems wrong with the conduction velocity data; 5 data points show a negative velocity. Are these antidromic spikes? Looking to Supplementary Table 1, some axon recordings are made at $\sim 50 \mu\text{m}$ distance from the soma. What kind of velocity are these data points reflecting? The authors need to explain how distance, initiation and temporal differences in onset were used to calculate a conduction velocity.

#2. Authors: "We observed no relationship between frequency and failure rate, shown in Supplementary Fig. 3, for either control axons or axons with swellings. This suggests that although axons fail more when driven to spike at high frequencies (e.g. Khaliq et al., 2005; Monsivais et al., 2005; Hirono et al., 2015), there appears to be no similar relationship between baseline firing rate and axonal failure, suggesting that baseline failure rate is set by a distinct mechanism to that which causes failures at high frequencies."

This important data could be part of a main figure. In the Discussion it is described as "surprising" which is puzzling. The spontaneous firing rate is between 20 and 80 Hz and not expected to generate failures in Purkinje cell axons given the published reliability of axons in this bandwidth. True, recording >1000 spontaneous spikes revealed a few more failures in axons with a swelling (Fig. 1). However, axonal failures really become significant at rates >250 Hz and a simple straightforward and well-controlled experiment is making paired recordings (from the same cell, not neighboring), inject current to elicit firing rates between 10 and 300 Hz at the soma and examine whether decreased failure rate is robustly different between axons with and without swellings. This experiment is fundamental to make a conclusion about the biophysical impact of swellings in cerebellar

information processing.

#4 Authors: "We state more clearly that we do not think that swellings have a direct effect on learning; rather, it likely occurs due to improved overall cerebellar performance arising from enhanced action potential propagation fidelity in the axon, and is thus likely indirect."

In the absence of conclusive data the authors are speculating about increased myelin thickness (see issue #1), about organelles changing the axial resistance (line 288, completely unsupported) and also about an intracellular mechanism (lines 116 and 295) that "may lead to specialized biochemical signaling which could contribute to the enhancement of action potential propagation that we have observed". How is intracellular IP3-receptor mediated signaling linked to voltage transfer?

Authors: "We chose to reword this because there are differences in the field about what constitutes a torpedo and what does not, and they have typically been used to describe disease-related swellings rather than those observed in healthy brains. We thus use the general term "axonal swelling" so that we are as clear as possible and do not confuse readers who have a different understanding of an axonal torpedo."

What is meant with "a different understanding of an axonal torpedo"? The premise seems they investigated torpedoes. In fact, the interchangeability between the two is spelled out in their core hypothesis in the abstract (line 8, p. 3) that we do not understand the biophysical impact of swellings (i.e. 'torpedoes') in neurodegenerative cases and that the authors now present evidence they ('torpedoes') facilitate transmission. Are these not the same structure? If not, this needs to be clarified and the neurodegeneration comments in the manuscript need to be revised.

Authors: "Furthermore, to better represent the different conditions, we have added bar graphs for each experiment allowing a comparison of axonal swellings after 3 hours. For this analysis, a Kruskal-Wallis H test (non-parametric one-way ANOVA) was used, followed by Mann-Whitney U tests with Bonferroni correction."

Panels i and j are showing same data as in panel h, but now re-plotted in a different manner and examined with a different statistical approach, as if this were to be an independent experiment. I'm afraid this is an unacceptable scientific practice. Panel h should show the post-hoc significance values for the row means. A time series analyzed with a repeated measures ANOVA has more power and contains more information. The same for the 3f, 4e, etc.

—

Reviewer #2 (Remarks to the Author):

The authors have been very responsive to my questions. I think this is a very interesting and important contribution that will force people to rethink these prominent swellings and their

Reviewer #3 (Remarks to the Author):

The authors have addressed most of my comments and significantly improved their manuscript. I have the following minor comment for further improvement.

(1) Line 317-320: "Indeed, we found that the number of axonal swellings correlates with motor performance on the last day.... ,albeit this correlation was weaker for some behaviour (data not shown)."

The authors should show the data as a supplementary figure.

(2) Line 311: "...that are then are relayed to the axon," should be read as "...that are then relayed to the axon,"

Reviewer #4 (Remarks to the Author):

The authors have partly addressed my comments.

Comment #2: The authors provide data on AP conduction velocity, but I don't understand what a negative speed of AP propagation is supposed to mean. The authors should show raw traces and indicate how the measurements were done.

Comment #3: I can believe that "that axonal failures lead to the formation of axonal swellings" but the evidence that these new swellings actually increase AP fidelity is not very strong. A successful before-after experiment could have nailed this issue, providing an internal control. I understand technical difficulty, so maybe authors should soften their conclusions in the absence of support by this kind of data.

Response to Reviewers

Reviewer #1 (Remarks to the Author)

In this revision the authors clarified some of the previous concerns by reanalyzing and re-plotting data together with textual changes. New data is shown in the supplement (although this would be better placed in the main results). While the topic still has broad interest and some aspects became clearer, this revision dampened my enthusiasm. The key question how a swelling alters Purkinje axon transmission fidelity remains unanswered and new errors have been introduced.

We thank the reviewer for their thoughtful and useful input on our manuscript. We have addressed the concerns raised by Reviewer 1 below and in our revised manuscript, and have moved some data from Supplementary figures to main figures.

#1A Regarding the EM, the claim (p. 14, line 261-264) of an increased myelin thickness underlying an enhanced action potential fidelity is not supported by the data. A g-ratio is not necessarily reflecting the myelin thickness. The g-ratio is the fraction of outer fiber diameter/axon diameter. Both can be different and not necessarily proportional. Showing a higher g-ratio thus leaves open various possibilities and Supplemental Figure 6 is required to interpret Figure 2. The authors need to show g-ratio as a function of linear axon diameter (removing also their error of the log₂ label). This reviewer plotted this graph based on the source data and the relationship suggests a continuum with larger axons having an increasingly thinner sheath. Such relationship is expected for the large diameter range between 1 and 10 μm (see e.g. Berthold and Rydmark, 1983). The only conclusion which can be drawn is that the swelling is not a synaptic terminal and the myelin sheath is relatively thinner but the largest difference is the axon core diameter (which is logic since the authors also selected for this parameter). There is a discrepancy with the functional data since if the myelin sheath is thicker across the entire internode the velocity should be higher (if measured over sufficiently large distances).

Given the reviewer's concerns about the validity of the interpretation of our myelin thickness measurements, we have now reported myelin thickness as function of axonal diameter, in addition to reporting the g-ratio this way, as the reviewer requested. The concept of g-ratio was proposed for axons of uniform diameter (Rushton 1951). Thus, it is not clear whether g-ratio is relevant for axons of variable diameter, like we observed with axonal swellings. For this reason, we have moved the g-ratio data to **Supplementary Fig. 6**, rather than a main figure. We explain this point in more detail in the Results (p.8 lines 8-10) and Discussion (p. 14 lines 9-11) sections. Our main conclusion from our EM work is that we see no synapses, as well as no accumulation of organelles in swellings compared to control axons, which we show in the main figure (new **Fig. 3**).

#1B. There is no increased velocity (Suppl Fig. 2d). However, something seems wrong with the conduction velocity data; 5 data points show a negative velocity. Are these antidromic spikes? Looking to Supplementary Table 1, some axon recordings are made at ~50 μm distance from the soma. What kind of velocity are these data points reflecting? The authors need to explain how distance, initiation and temporal differences in onset were used to calculate a conduction velocity.

Purkinje cell axons are believed to initiate their action potentials in the axon initial segment (AIS), which propagate from this location both down the axon and into the soma (Palmer et al., 2010). It has been previously shown that action potentials can thus appear first at an axonal recording site, and later at a somatic recording site (Palmer et al., 2010). Indeed, Palmer and colleagues show that with three different techniques: extracellular recordings, voltage-sensitive dye imaging, and cell-attached recordings, action potentials often appear in the axon before the soma when recording within the first ~150 μm of the axon. This is well within the distances that we have examined, and the negative delays we observe lie within the range of negative axosomatic delays reported by Palmer and colleagues, meaning that our findings are consistent with published work. Given this, the concept of velocity is complex since the action potential is not propagating in one direction with our recording configuration, but rather from the AIS both towards the cell body and down the axon. We now report axosomatic delay rather than velocity to reduce confusion (consistent with the published work of Palmer and colleagues; **Fig. 1e** and **Fig. 2d**).

#2. Authors: “We observed no relationship between frequency and failure rate, shown in Supplementary Fig. 3, for either control axons or axons with swellings. This suggests that although axons fail more when driven to spike at high frequencies (e.g. Khaliq et al., 2005; Monsivais et al., 2005; Hirono et al., 2015), there appears to be no similar relationship between baseline firing rate and axonal failure, suggesting that baseline failure rate is set by a distinct mechanism to that which causes failures at high frequencies.”

This important data could be part of a main figure. In the Discussion it is described as “surprising” which is puzzling. The spontaneous firing rate is between 20 and 80 Hz and not expected to generate failures in Purkinje cell axons given the published reliability of axons in this bandwidth. True, recording >1000 spontaneous spikes revealed a few more failures in axons with a swelling (Fig. 1). However, axonal failures really become significant at rates >250 Hz and a simple straightforward and well-controlled experiment is making paired recordings (from the same cell, not neighboring), inject current to elicit firing rates between 10 and 300 Hz at the soma and examine whether decreased failure rate is robustly different between axons with and without swellings. This experiment is fundamental to make a conclusion about the biophysical impact of swellings in cerebellar information processing.

We thank the reviewer for their suggestion. The spontaneous firing rate data is now included in **Fig. 1f**, and we no longer report it as “surprising.” Furthermore, we have done the experiments suggested by the reviewer to make paired recordings combining whole-cell somatic recordings

with axonal recordings from the same cell. We have done this in axons both with and without swellings. We observe swellings in developing mice, and have done our electrophysiological recordings in younger mice (P9-P16). As previously reported (McKay and Turner, 2005), we find that Purkinje cells at this age have a range of different maximal firing frequencies, and we were not typically able to drive them at high frequencies, so do not report frequencies up to 300 Hz as the reviewer suggests (rather, up to 200 Hz). Nonetheless, we observe that control cells without swellings exhibited reduced axonal propagation success near their maximal firing frequencies, and that Purkinje cells with axonal swellings showed significantly enhanced propagation success at higher frequencies. These results have been included in a new figure (new **Fig. 2**), which we feel adds additional support to our observations that axonal swellings enhance axonal propagation.

#4 Authors: "We state more clearly that we do not think that swellings have a direct effect on learning; rather, it likely occurs due to improved overall cerebellar performance arising from enhanced action potential propagation fidelity in the axon, and is thus likely indirect."

In the absence of conclusive data the authors are speculating about increased myelin thickness (see issue #1), about organelles changing the axial resistance (line 288, completely unsupported) and also about an intracellular mechanism (lines 116 and 295) that "may lead to specialized biochemical signaling which could contribute to the enhancement of action potential propagation that we have observed". How is intracellular IP3-receptor mediated signaling linked to voltage transfer?

We do not know if IP3 receptor mediated signaling is linked to voltage transfer in the axon, and have rewritten the Discussion, omitting this admittedly rather speculative section in our Discussion entirely.

Authors: "We chose to reword this because there are differences in the field about what constitutes a torpedo and what does not, and they have typically been used to describe disease-related swellings rather than those observed in healthy brains. We thus use the general term "axonal swelling" so that we are as clear as possible and do not confuse readers who have a different understanding of an axonal torpedo."

What is meant with "a different understanding of an axonal torpedo"? The premise seems they investigated torpedoes. In fact, the interchangeability between the two is spelled out in their core hypothesis in the abstract (line 8, p. 3) that we do not understand the biophysical impact of swellings (i.e. 'torpedoes') in neurodegenerative cases and that the authors now present evidence they ('torpedoes') facilitate transmission. Are these not the same structure? If not, this needs to be clarified and the neurodegeneration comments in the manuscript need to be revised.

There is considerable debate in the field of neurodegenerative diseases about what constitutes a torpedo and what does not, whether swellings distal on the axon are the same as those near the

soma. These structures have historically been under-studied, meaning that a definitive answer is lacking. We do not know if swellings observed in neurodegenerative diseases are the same structures as axonal swellings that we have observed in developing and young adult mice, although given their similarities at the light microscopic level, this is a possibility. Indeed, it is unknown whether axonal swellings observed in different disease are all the same or not, since there are differences reported at the EM level in swelling composition in different diseases, which could suggest that there are different populations of swellings even across diseases, or even within a disease. Because of this confusion, we have chosen to not call the swellings we are observing “torpedoes,” in contrast to our 2016 publication (Ljungberg et al., 2016), where we do. We have instead chosen “axonal swelling” because it is a neutral, descriptive term. We have revised the section on neurodegenerative diseases in the Introduction to make it clearer that the field lacks knowledge about whether these structures in disease are the same as in development. Furthermore, we raise this issue in the Discussion section as well (page 16, lines 16-23).

We have rewritten the text to make it clearer that it is not currently known whether swellings in development and swellings in disease are the same structures or serve the same function (e.g. page 3, lines 11-12). We have also rewritten the abstract to make clearer that computational modeling predicts impaired axonal function for swellings, and that this claim does not arise simply from their presence in neurodegenerative disease (page 2, lines 6-7).

Authors: “Furthermore, to better represent the different conditions, we have added bar graphs for each experiment allowing a comparison of axonal swellings after 3 hours. For this analysis, a Kruskal-Wallis H test (non-parametric one-way ANOVA) was used, followed by Mann-Whitney U tests with Bonferroni correction.”

Panels i and j are showing same data as in panel h, but now re-plotted in a different manner and examined with a different statistical approach, as if this were to be an independent experiment. I’m afraid this is an unacceptable scientific practice. Panel h should show the post-hoc significance values for the row means. A time series analyzed with a repeated measures ANOVA has more power and contains more information. The same for the 3f, 4e, etc.

We thank the reviewer for their input and have revised the figures based on these suggestions. The post-hoc significance values are indeed reported after the repeated measures ANOVA. The new Supplemental Table 2 now shows all the significant for the Repeated Measures ANOVA.

Reviewer #2 (Remarks to the Author):

The authors have been very responsive to my questions. I think this is a very interesting and important contribution that will force people to rethink these prominent swellings and their

We thank the reviewer for their thoughtful and useful input on our manuscript.

Reviewer #3 (Remarks to the Author):

The authors have addressed most of my comments and significantly improved their manuscript. I have the following minor comment for further improvement.

We thank the reviewer for their thoughtful and useful input on our manuscript.

(1) Line 317-320: "Indeed, we found that the number of axonal swellings correlates with motor performance on the last day.... ,albeit this correlation was weaker for some behaviour (data not shown)."

The authors should show the data as a supplementary figure.

We have included this figure as **Supplementary Fig. 11**.

(2) Line 311: "...that are then are relayed to the axon," should be read as "...that are then relayed to the axon,"

Thanks for catching this typo, we have fixed it.

Reviewer #4 (Remarks to the Author):

The authors have partly addressed my comments.

Comment #2: The authors provide data on AP conduction velocity, but I don't understand what a negative speed of AP propagation is supposed to mean. The authors should show raw traces and indicate how the measurements were done.

We thank the reviewer for thoughtful and useful input on our manuscript. We have added in an illustration of how we measure this information, as suggested, in **Supplemental Fig. 2**. We ask the reviewer to consider the response to Reviewer 1 Question 1B above. Briefly, our findings are in line with what has been reported in the literature (e.g. Palmer et al., 2010), since action potentials are initiated in the axon initial segment, and thus what we are measuring is its propagation in two directions to be detected by two electrodes (somatic and axonal). It is thus possible for an action potential to propagate to nearby sites on the axon before the soma, and this has been reported for Purkinje cell axons at recording locations within ~150 μm of the soma (Palmer et al., 2010). Thus, our findings are consistent with published reports. We have revised how we report this data, and now show the axosomatic delay between the recording electrodes rather than speed/velocity.

Comment #3: I can believe that "that axonal failures lead to the formation of axonal swellings" but the evidence that these new swellings actually increase AP fidelity is not very strong. A

successful before-after experiment could have nailed this issue, providing an internal control. I understand technical difficulty, so maybe authors should soften their conclusions in the absence of support by this kind of data.

We appreciate the Reviewer's understanding of the technical challenges of the proposed experiments. While we agree that before and after experiments would be very conclusive, these are so technically challenging to make them unattainable in our hands (we are unable to hold axonal recordings for the 1.5-2 hours that would be necessary for these experiments). We have, however, included additional data looking at frequency-related axonal failures, which has been observed in the past when Purkinje cells are driven to fire at very high frequencies (e.g. Khaliq et al., 2005, Monsivais et al., 2005, Hirono et al., 2015). We observe something similar in our young control axons, and find that axonal swellings enhance axonal propagation. We believe that this further piece of evidence adds additional strength to our claims. However, we have also softened our conclusions as the reviewer suggests in several instances throughout the text (e.g. p. 17 line 8, 11).

Reviewer #1 (Remarks to the Author):

In this revision the authors provide new data and analyses including experimentally controlled firing rate changes of the Purkinje cell revealing more failures in axons containing swellings. These new results (Fig. 2) are compelling and strengthen the idea that swellings in young adult Purkinje cell axons improve conduction fidelity for action potentials. Overall, the findings are intriguing and an important contribution to our understanding of the cell biology and function of Purkinje axon torpedoes (or swellings). I have a few minor comments.

- Please indicate in Figure 2 the steps with which the firing frequency was increasing.
- Lines 140–150 and 288–294. The line of argumentation in these paragraphs is highly convoluted. As was previously discussed in the rebuttal, based on the EM the authors find evidence that the swellings are local changes in the diameter of the Purkinje axon, without changes in the myelin sheath thickness. Why bringing up Rushton? The mathematical equations solve how conduction velocity compares to the relationships between myelin thickness and axon diameter, based on populations of myelinated axons. The theory is, however, not applicable to the analysis of highly localized (and rapidly evolving) diameter changes. Writing 'at odds' is illogical and will confuse the readers. A relevant point to discuss possible mechanistic explanations how a local axon diameter increase improves spike fidelity.
- Line 167 change 'to confirm that' into 'to test whether'.
- Line 180. 'wondered whether this selectivity arose from morphological differences between axons'. It reads as if the authors forgot they showed in Figure 3 that in the cerebellum Purkinje cell axons with and without swellings have similar diameters (Supplementary Fig. 4).
- Line 227. The role of the Monte Carlo modelling is unclear. The authors are not drawing conclusions from the results and neither put these findings into context. Please clarify and integrate with the behavioral analyses.

Reviewer #4 (Remarks to the Author):

I have no further comments, the study should be published without further delay.

Response to Reviewers

Reviewer #1 (Remarks to the Author):

In this revision the authors provide new data and analyses including experimentally controlled firing rate changes of the Purkinje cell revealing more failures in axons containing swellings. These new results (Fig. 2) are compelling and strengthen the idea that swellings in young adult Purkinje cell axons improve conduction fidelity for action potentials. Overall, the findings are intriguing and an important contribution to our understanding of the cell biology and function of Purkinje axon torpedoes (or swellings). I have a few minor comments.

We thank the reviewer for their positive comments about the new data that is included.

– Please indicate in Figure 2 the steps with which the firing frequency was increasing.

We have updated Fig. 2 to indicate the specific current steps for the recordings shown in the legends, and the range of current steps we used is shown as an inset in Fig 2a.

– Lines 140–150 and 288–294. The line of argumentation in these paragraphs is highly convoluted. As was previously discussed in the rebuttal, based on the EM the authors find evidence that the swellings are local changes in the diameter of the Purkinje axon, without changes in the myelin sheath thickness. Why bringing up Rushton? The mathematical equations solve how conduction velocity compares to the relationships between myelin thickness and axon diameter, based on populations of myelinated axons. The theory is, however, not applicable to the analysis of highly localized (and rapidly evolving) diameter changes. Writing 'at odds' is illogical and will confuse the readers. A relevant point to discuss possible mechanistic explanations how a local axon diameter increase improves spike fidelity.

We thank the reviewer for their suggestions to improve the clarity of these passages. We have rewritten them, and have included as a discussion point the importance of understanding mechanistic explanations of how a local axon diameter increase could improve spike fidelity in the Discussion section.

– Line 167 change 'to confirm that' into 'to test whether'.

We have updated this line to reflect the reviewer's suggestion.

– Line 180. ‘wondered whether this selectivity arose from morphological differences between axons’. It reads as if the authors forgot they showed in Figure 3 that in the cerebellum Purkinje cell axons with and without swellings have similar diameters (Supplementary Fig. 4).

We thank the reviewer for his/her input, and have altered this text to make the logic of this experiment clearer.

– Line 227. The role of the Monte Carlo modelling is unclear. The authors are not drawing conclusions from the results and neither put these findings into context. Please clarify and integrate with the behavioral analyses.

We thank the reviewer for his/her comments, and have made the suggested edits to the Results section and the figure legend.

Reviewer #4 (Remarks to the Author):

I have no further comments, the study should be published without further delay.

We thank the reviewer for their positive comments about the manuscript.